Manuscript prepared for Atmos. Chem. Phys.
with version 2014/09/16 7.15 Copernicus papers of the LATEX class copernicus.cls.
Date: 10 August 2020

# Effects of global ship emissions on European air pollution levels

Jan Eiof Jonson[1], Michael Gauss[1], Michael Schulz[1], Jukka-Pekka Jalkanen[2], and Hilde Fagerli[1]

[1]Norwegian Meteorological Institute, Oslo, Norway
[2]Finnish Meteorological Institute, Helsinki, Finland

*Correspondence to:* Jan Eiof Jonson (j.e.jonson@met.no)

**Abstract.**

Ship emissions constitute a large, and so far poorly regulated, source of air pollution. Emissions are mainly clustered along major ship routes, both in open seas and close to densely populated shorelines. Major air pollutants emitted include sulfur dioxide, $NO_x$ and primary particles. Sulfur and $NO_x$ are both major contributors to the formation of secondary fine particles ($PM_{2.5}$) and to acidification and eutrophication. In addition, $NO_x$ is a major precursor for ground-level ozone. In this paper we quantify the contributions from international shipping to European air pollution levels and depositions.

This study is based on global and regional model calculations. The model runs are made with meteorology and emission data representative for year 2017, after the tightening of the SECA (Sulfur Emission Control Area) regulations in 2015, but before the global sulfur cap that entered into force in 2020. The ship emissions have been derived using ship positioning data. We have also made model runs reducing sulfur emissions by 80% corresponding to the 2020 requirements. This study is based on model sensitivity studies perturbing emissions from different sea areas: the Northern European SECA in the North Sea and the Baltic Sea, the Mediterranean Sea and the Black Sea, the Atlantic Ocean close to Europe, shipping in the rest of the world, and finally all global ship emissions together. Sensitivity studies have also been made setting lower bounds on the effects of ship plumes on ozone formation.

Both global and regional scale calculations show that for $PM_{2.5}$ and depositions of oxidised nitrogen and sulfur, the effects of ship emissions are much larger when emissions occur close to the shore than at open seas. In many coastal countries calculations show that shipping is responsible for 10% or more of the controllable $PM_{2.5}$ concentrations and depositions of oxidised nitrogen and sulfur. With few exceptions the results from the global and regional calculations are similar. Our calculations

show that substantial reductions in the contributions from ship emissions to $PM_{2.5}$ concentrations and to depositions of sulfur can be expected in European coastal regions as a result of the implementation of a 0.5% worldwide limit of the sulfur content in marine fuels from 2020. For countries bordering the North Sea and Baltic Sea SECA, low sulfur emissions already resulted in marked reductions in $PM_{2.5}$ from shipping before 2020.

For ozone the lifetime in the atmosphere is much longer than for $PM_{2.5}$, and the potential for ozone formation is much larger in otherwise pristine environments. We calculate considerable contributions from open sea shipping. As a result we find that the largest contributions to ozone in several regions and countries in Europe are from sea areas well outside European waters.

## 1 Introduction

As shown by both model calculations and measurements, concentrations of almost all air pollutants have decreased throughout most of Europe since 1990 Colette et al. (2016, 2017). Over the same time span, depositions of eutrophying and in particular acidifying species have also decreased (Theobald et al., 2019). These trends are, with the partial exception of ground level ozone, almost entirely driven by reductions in European land based emissions (Colette et al., 2016). Since year 2000, European emission trends are diverse, with general downward trends in Western European countries and upward trends in Eastern Europe (Gaisbauer et al., 2019). The latter upward trends are largely driven by an economic recovery in former Soviet Union states.

Emissions from international shipping to air, relevant in the context of air pollution and depositions in Europe, include PPM (primary particulate matter), sulfur, $NO_x$, CO, and NMVOC (Non-Methane Volatile Organic Carbon). Trends in emissions from shipping are less certain than for land based emissions, and differ depending on species and sea area. In general emissions from shipping have changed less than land based emissions in Western Europe (Gaisbauer et al., 2019), and, as a result, the relative contributions of ship emissions to air pollution and depositions in western parts of Europe have increased. One notable exception is the SECA (Sulfur Emission Control Area) regions in the Baltic Sea and the North Sea, where sulfur emissions have dropped by more than an order of magnitude in the last decade. In the SECAs the maximum allowed sulfur content in fuels, and consequently the emissions from shipping, has been reduced in several steps with the latest, and most significant, measure implemented from January 2015 reducing the maximum allowed sulfur content in marine fuels from 1% to 0.1% (IMO, 2008). Fuels with higher sulfur content may be used in combination with technology reducing sulfur emission to levels equivalent to the use of low-sulfur fuels. In addition the EU sulfur directive requires ships to use fuel with 0.1 % sulfur in EU harbour areas. From 2020 a global cap on sulfur content in marine fuels of 0.5% has been implemented as opposed to an average of about 2.5 % prior to 2020.

The global effects of international shipping on air pollution and depositions have already been identified in several papers (Corbett et al., 2007; Endresen et al., 2003; Eyring et al., 2007; Sofiev et al., 2018). In a global model calculation, Jonson et al. (2018a) found that a large portion of the anthropogenic contributions to $PM_{2.5}$ and depositions of sulfur and nitrogen in European coastal regions can be attributed to ship emissions in nearby sea areas. For boundary layer ozone the same study showed a mixed result, with overall percentage contributions to ozone of antropogenic origin of more than 20% in several Mediterranean countries, and negative contributions in some countries bordering the North Sea caused by ozone titration. In Jonson et al. (2018b) the effects of pollution from other continents, including also the effects of international shipping on European air pollution, were investigated within the framework of TF_HTAP2 (Task Force on Hemispheric Transport of Air Pollution, phase II) (http://www.htap.org/, last accessed 7 July 2020. These calculations indicated that more than 10% of the ozone in Europe of anthropogenic origin can be attributed to international shipping. The percentage contributions were similar for both annually averaged ozone and for ozone indicators such as SOMO35[1] and $POD_1$ (deciduous) forest[2].

In Karl et al. (2019) the EMEP model, along with two other models, was applied in a regional study on the effects of ship emissions in the Baltic Sea region using meteorology and emissions for year 2012. In that study the average contribution of ships to levels of $PM_{2.5}$ ranged from 4.15 to 6.5 % in the entire Baltic Sea region, and from 3.15 to 5.7 % in the coastal land areas. In addition the model results were compared to measurements in the region. Jonson et al. (2019) found that the implementation of stricter SECA regulations in the Baltic Sea and the North Sea from 2015 resulted in marked reductions $PM_{2.5}$ levels in the Baltic Sea region. In a companion paper using the same data Barregård et al. (2019) estimated that the number of premature deaths from shipping dropped by one third between 2014 and 2016.

With ship emissions representative for year 2013, Lv et al. (2018) calculated contributions from ship emissions to $PM_{2.5}$ concentrations of up to 5.2 $\mu gm^{-3}$ in coastal regions of China, higher than in European coastal regions. Since 2013 emission controls have been imposed in China in several steps, limiting the fuel sulfur content in marine fuels to 0.5% in several Chinese ports and territorial waters.

In this paper we study the effects of global international shipping further by performing a series of scenario calculations perturbing ship emissions, both globally and from individual sea areas, to attribute the effects of ship emissions on European countries from different sea areas. We have limited the study to air concentrations of $PM_{2.5}$ and ozone, and to depositions of oxidised nitrogen and sulfur. The calculations are made with meteorology and emissions for year 2017, but calculations are also

---

[1]SOMO35 (Sum of Ozone Means Over 35 ppb) is the indicator for health impact assessment recommended by WHO. It is defined as the yearly sum of the daily maximum of the running 8-hour running average of ozone above 35 ppb.

[2]$POD_1$ (Phyto-toxic Ozone Dose for deciduous forests) is the accumulated stomatal ozone flux over a threshold Y integrated from the start to the end of the growing season. For deciduous forests, the critical level of 4 mmol m$^{-2}$ is exceeded in most of Europe, indicating a risk of ozone damage to forests. See Mills et al. (2011a, b) for further description of this metric.

made for 2020 and beyond, by scaling sulfur emissions outside the North Sea and Baltic Sea SECAs by 0.5/2.5 (a decrease in the sulfur content in marine fuels from about 2.5% to 0.5%), reflecting the expected reductions in sulfur emissions following the CAP2020 regulations implemented in 2020, see http://www.imo.org/en/mediacentre/hottopics/pages/sulphur-2020.aspx, last accessed 7 July 2020.

The global model calculations are compared to the regional scale source receptor calculations, also for year 2017, included in the latest EMEP report (EMEP Status Report 1/2019, 2019).

Finally, sensitivity tests have been made to give bounds for the effect of chemistry within exhaust plumes. In pristine environments, pollutant concentrations can be orders of magnitude higher within ship plumes than in their surroundings, whereas in the model these emissions plumes are instantly
diluted into a large grid volume. Ignoring the chemistry within the plumes can potentially result in an overestimation of ozone.

## 2    Model description

Concentrations of air pollutants and depositions of sulfur and nitrogen have been calculated with the EMEP MSC-W model version rv4.34 (hereafter 'EMEP model') on a global model domain
with a 0.5° x 0.5° longitude-latitude resolution. The EMEP model is a comprehensive air quality model which has been used extensively during the last four decades for air pollution research and to underpin international air quality legislation. It takes into account processes of emissions, advection, turbulent diffusion, chemical transformations, and wet and dry depositions. The calculations of dry depositions are made separately for each sub-grid land-cover classification. These sub-grid estimates
are aggregated to provide output deposition estimates for broader ecosystem categories as deciduous and coniferous forests. A detailed description of the EMEP model can be found in Simpson et al. (2012) with later model updates being described in Simpson et al. (2019) and references therein. The EMEP model is available as Open Source (see https://github.com/metno/emep-ctm), last accessed 7 July 2020.

For comparison we also include results from the regional model calculations included in the latest EMEP report (EMEP Status Report 1/2019, 2019) covering the geographical area between 30°N–82° and 30°W–90°E on a 0.3° x 0.2° longitude-latitude resolution. Both the global and regional regional calculations have been made using 2017 meteorological input data and 2017 emissions. The meteorological input data are from the European Centre for Medium-Range Weather Forecasts
(ECMWF) based on the CY40R1 version of their IFS (Integrated Forecast System) model.

### 2.1   Model evaluation and comparisons to other models

The EMEP model is under continuous development, and undergoes extensive evaluation against measurements every year as part of the EMEP status reports, see Gauss et al. (2017, 2018, 2019) for evaluations of the latest emission years available, 2015, 2016, and 2017. The model is also evaluated

daily and openly within the Copernicus Atmosphere Monitoring Service, where it is used operationally for regional air quality forecasts and analyses (see https://www.regional.atmosphere.copernicus.eu/), last accessed 7 July 2020. In addition, the EMEP regional model has successfully participated in model inter-comparisons and model evaluations in a number of peer-reviewed publications (Colette et al., 2011, 2012; Angelbratt et al., 2011; Dore et al., 2015) In Vivanco et al. (2018), depositions of sulfur

and nitrogen species in Europe calculated by 14 regional models were evaluated against measurements showing good results for the EMEP model. In global mode the model has also participated in a number of model inter-comparisons and model evaluations (Stjern et al., 2016; Tan et al., 2018; Liang et al., 2018; Jonson et al., 2018a). At least for background sites, the performance is comparable for regional and global model applications.

In Karl et al. (2019) the EMEP model, the SILAM model, and the CMAQ model were compared to measurements, and in terms of calculated effects of ship emissions in the Baltic Sea. For $PM_{2.5}$, both the CMAQ and the EMEP models had a slightly negative bias, whereas the SILAM model had virtually no bias. Even so, the SILAM model calculated a slightly lower contribution from Baltic Sea shipping compared to the other two models. All three models overpredicted ozone for urban

measurement sites. The EMEP model also had a moderate positive bias at rural sites. The EMEP model calculated less ozone titration in the shipping lanes, most likely as a result of its coarser model resolution compared to the other two models. Over land, all three models calculated small increases in ozone due to ship emissions. In Jonson et al. (2019) the importance of ship emissions was demonstrated by comparing model results and Baltic Sea coastal measurements for 2016 for

$PM_{2.5}$, $SO_2$, $NO_2$, and ozone. Also here, the EMEP model had a negative bias for $PM_{2.5}$. $NO_2$ concentrations were severely underestimated when ship emissions were set to zero, illustrating the importance of ship emissions in the real atmosphere. Likewise 2016 $SO_2$ concentrations were strongly overestimated when using 2014 emissions.

### 2.2   Emissions

For the global calculations land-based emissions have been provided by the International Institute for Applied Systems Analysis (IIASA) within the European FP7 project ECLIPSE (http://www.iiasa.ac.at/web/home/research/researchPrograms/air/ECLIPSEv5.html, last accessed / July 2020). In this study we use ECLIPSE version 6a (hereafter referred to as 'ECLIPSEv6a'), which is a global emission data set on 0.5 x 0.5 degree resolution and is widely used by the scientific community. Some

of the methods used in ECLIPSE are described in the recent publication of Höglund-Isaksson et al. (2020). Historical data rely on statistical data (until 2015) for energy from the International Energy Agency (IEA), agricultural data from the United Nations Food and Agriculture Organisation (FAO), the International Fertiliser Association (IFA), and additional data for mineral industries from United States Geological Survey (USGS), and numerous additional sources for informal industries (e.g.,

brick making), waste, etc. Current baseline projections rely on the New Policies Scenario (NPS) from

the World Energy Outlook 2018 of IEA (IEA, 2018) FAO projections, and for EU agriculture also on the European-wide farmtype model in CAPRI (Common Agricultural Policy Regional Impact). ECLIPSEv6a emissions are available in 5-year intervals from 2005 onward. In this study the emissions are interpolated to 2017.

The land-based emissions used in the regional model calculations are described in Gaisbauer et al. (2019) and are mainly based on the officially reported data from the countries. In Table 1 these officially reported emissions are listed aggregated for the EU27 countries compared to the ECLIPSEv6a emissions. Differences are of similar magnitude for the individual EU countries. The most significant difference is for sulfur, where the ECLIPSEv6a emissions are of the order of 15%

higher than those reported to EMEP.

    Ship emission data sets used in both the global and regional model calculations are originally from the Finish Meteorological Institute, based on AIS data processed in the STEAM model (Johansson et al., 2017) and downloaded from the ECCAD database (https://eccad.aeris-data.fr/), last visited 7 July 2020. Ship emissions of various species, based on the global data set, are listed in Table 1

separately for the Baltic Sea, the North Sea (including the English Channel), the Mediterranean Sea, and the Black Sea. In addition emissions are listed for the remaining Atlantic area outside Europe, but bounded by 30 – 82 degrees north and 30 degrees west to 90 degrees east corresponding to the "Northeast Atlantic Ocean" also included in the regional calculations. These three sea areas are depicted in Figure 1. Finally emissions are also listed for the total global sea area. Annual ship

emissions used in the regional model calculations are based on the same source (Gaisbauer et al., 2019). Even so, total ship emissions in the sea areas as used in the global calculations are somewhat higher than in the regional calculations (see EMEP Status Report 1/2019 (2019), appendix B for comparison).

    In the FMI emission data all PPM emissions are assumed to be emitted as $PM_{2.5}$. Emissions from

leisure boats are not included. In a separate study Johansson et al. (2020) have quantified the emissions from leisure boats in the Baltic Sea only. Compared to emissions from the commercial fleet these emissions were insignificant for $NO_x$ and PPMs. However, in regard to emissions of NMVOC the study concluded that these can be significantly larger from leisure boats than from registered vessels in the Baltic Sea, especially during summer (about 500% larger). However, as shown in Table 1, the

NMVOC to $NO_x$ ratio is close to 1 for land based emissions, but very low for ship emissions.

**2.3   Definition of the model sensitivity tests**

In order to calculate the effects of ship emissions on air pollution and depositions in Europe we use a similar approach as in the SR (Source Receptor) calculations within the EMEP programme (see EMEP Status Report 1/2019 (2019) appendix C as the latest example). We reduce the emissions by

15% in the sea areas in order to make the results comparable to the regional EMEP SR results. Both the global and regional model runs are made for a full calendar year (2017). As some of the species

have a long lifetime in the atmosphere (one month or more), the global model runs are preceded by a 5-month spin-up. But for model runs perturbing only a limited sea area, the spin-up from the Base model run is used (see Table 2). Whereas in the regional EMEP SR calculations emissions of different species are reduced in separate perturbation runs, we in the global runs reduce the emissions of all species simultaneously in the same perturbation run, reducing the number of model runs to one for each of the model scenarios listed in Table 2. We have combined the North Sea and the Baltic Sea into one scenario run because they are both designated as SECA areas. Likewise we have combined the Mediterranean Sea and the Black Sea. The sea areas are shown in Figure 1. ROW (Rest Of World) are all sea areas not included in the sea areas listed above. We have also made additional model runs with sulfur emissions from ships reduced to CAP2020 levels.

In the interpretation of the model results below we let the difference between the Base_2017 and the SR_AllAnt model runs (see Table 2) represent 100% of the effects of all anthropogenic, and thus controllable, global emissions. Similarly we calculate the contributions from global shipping as a whole, or from shipping in a specific area, by subtracting the scenario run for shipping as a whole or from a specific sea area from the Base model run. In this way we can relate the effects of ship emissions in different regions to the total anthropogenic contribution. Even though not linear, this is a widely used approach that, in addition to in the EMEP reporting, was also taken in the TF_HTAP phase II modelling exercise (see workplan under http://www.htap.org/, last accessed 7 July 2020). Reducing the emissions by a different percentage would give slightly different results depending on the species and location. The choice of 15% is partly political as reductions of this magnitude are achievable within a timeframe of a few years and at the same time they give a large enough signal when processing the model output.

For all depositions and air concentrations except ozone (and ozone metrics) we add up the SR runs for the individual sea areas (SR_BALNOS, SR_MEDBLS, SR_ATL, and SR_ROW) and compare with the SR_AllSh emission perturbation providing a measure of the linearity in the calculations.

In the model calculations described above, the ship emissions are instantly diluted throughout the model grid cells in which the emissions occur. Previous studies (Vinken et al., 2011; Huszar et al., 2010) have shown that this can lead to an overestimation of the ozone formation, in particular in sea areas where $NO_x$ concentrations are otherwise low. The EMEP model has an option for splitting 50% of the $NO_x$ emissions from shipping into a pseudo-species "ShipNOx", and the other half emitted as NO and $NO_2$ as in the Base model runs, see Simpson et al. (2015). ShipNOx deposits as $NO_2$, but undergoes simple atmospheric reactions:

$$\text{ShipNOx} + \text{OH} \quad \Rightarrow \quad \text{HNO}_3 \quad [\text{R1}]$$
$$\text{ShipNOx} \quad \Rightarrow \quad \text{HNO}_3 \quad [\text{R2}]$$

Reaction R1 proceeds with the same rate as the normal $NO_2$ + OH reaction, thus proceeding faster in daylight and in high OH areas. Reaction R2 provides a minimum half-life of about 6 hours, loosely

based upon results shown in Vinken et al. (2011). We have repeated the calculations for the scenarios listed above with the ShipNOx reactions included. We then assume that the calculations with and

without the ShipNOx split represent a lower and an upper limit of the effects of $NO_x$ emissions from shipping on the formation of ozone both globally and in the individual sea areas.

## 3 Model results

In this section we show the calculated effects of all global ship emissions, and the effects of emissions from separate sea areas as defined in the separate scenarios in Section 2.3 . For ozone we also include

a discussion on the effects of the ShipNOx split, and for $PM_{2.5}$ we include the effects of the CAP2020 regulations.

Below we include the model results from all ship emissions, and from ship emissions in separate sea areas based on the model scenarios listed in Table 2. For the calculations perturbing the emissions in separate sea areas, the total effect in a receptor area will then be the sum of contributions

from all the individual sea areas. This sum will be a combination of the emission and chemical production/destruction of the species within the source sea area, and production/destruction of the species elsewhere (including the receptor region). Similar positive and negative contributions were also identified in the TF_HTAP2 model experiment, as exemplified by the results in Jonson et al. (2018b) and in the EMEP source receptor calculations, as exemplified by EMEP Status Report 1/2019

(2019), appendix C. Thus, for example, reductions in the receptor area can be caused by chemical reactions that only occur in the source area (e.g. ozone titration), followed by transport of a smaller amount of the species (e.g. ozone) into the target area.

### 3.1 $PM_{2.5}$

Figure 2 shows the global concentration of $PM_{2.5}$ (a) and the contributions from global shipping (b).

Globally the highest concentrations are calculated over parts of Asia and North Africa. In Europe high concentrations are calculated in several locations with the highest concentrations in the Po Valley in Italy. The largest contributions from shipping are mainly calculated in and around the major ship tracks. In Figure 3 we show to what extent ship emissions from different sea areas contribute to the European $PM_{2.5}$ concentrations seasonally. From all sea areas the largest effects are calculated in

nearby countries/regions. Ship emissions generally peak in summer, but the seasonal variations in emissions are not large, and far from large enough to explain the seasonal variations in concentrations seen in Figure 3. The main sources for particles and particle formation from shipping are $NO_2$ and sulfur (of which more than 95% is emitted as $SO_2$ in the gas phase, and the rest as sulfate particles). In addition ash, EC (Elemental Carbon), and OC (Organic Carbon) are assumed emitted as primary

particles. The main oxidation paths for $SO_2$ are the OH reaction in the gas phase and in-cloud oxidation (mainly with $H_2O_2$). Both these oxidants have a clear summer maximum, contributing to a

summer maximum also for sulfate. In sea areas outside the SECAs sulfate makes up 50 to 80% of the $PM_{2.5}$, dry mass (Figure 8a), explaining the summer maximum in $PM_{2.5}$ concentrations in most sea areas.

The second largest fraction is nitrate (Figure 8b). $NO_2$ is oxidised to gaseous $HNO_3$. $HNO_3$ can then react with sea salt forming particulate sodium nitrate, but these particles are in general large, not contributing to $PM_{2.5}$. However, in the presence of ammonia the formation of ammonium nitrate particles can be a lot faster. The latter reaction requires a surplus of $NH_3$ over sulfate. Ammonia is mainly emitted from agriculture with a seasonal maximum in spring. The nitrate fraction from

shipping is large in the SECA sea areas where sulfur emissions are very low, and particularly high over land.

    Although ammonia is not emitted from ships, nitrate and sulfate from ships increase the formation of ammonium sulfate and ammonium nitrate so that the ammonium makes up 20 to 30% of the $PM_{2.5}$ over parts of the European continent (Figure 8d). However, as shown in Figure 3 $PM_{2.5}$

concentrations over land are very low except for the coastal zones.

    In the SECA sea areas we calculate that as much as 20 to 30% of the$PM_{2.5}$ from shipping comes from the primary particles ash, EC, and OC (Figure 8c). .

    The effects of the emissions from individual sea areas on $PM_{2.5}$ discussed below are based on 2017 ship emissions. The effects of the CAP2020 global reductions in sulfur emissions from ships

are described in Section 4.

### 3.1.1   Contributions from the North Sea and the Baltic Sea

For countries/regions bordering the North Sea and the Baltic Sea (Figure 3 a,b,c,d) $PM_{2.5}$ from local shipping peaks in spring. Following the implementation of the stricter SECA regulations from 2015, sulfur emissions are low (see Table 1). In particular the southwestern parts of this sea area are close to

some of the highest ammonia emission regions in Europe. The main source of particles from shipping is $NO_2$ through the formation of nitrate, predominantly ammonium nitrate. The spring maximum in $PM_{2.5}$ from the North Sea and the Baltic Sea shipping is caused by the interaction with ammonia emissions, mainly from agriculture, peaking in spring.

### 3.1.2   Contributions from the Northeast Atlantic Ocean

The largest contributions to $PM_{2.5}$ concentrations in Europe from shipping in the Northeast Atlantic (see Figure 3 e,f,g,h) are calculated for the regions bordering the ship track in and out of the Mediterranean through Gibraltar, extending north to the English Channel. As this region is outside the SECA, sulfur emissions are high, and a major constituent in $PM_{2.5}$ from shipping is sulfur, emitted mainly as gaseous $SO_2$ and then oxidised to sulfate. The summer maximum in the contributions from

the Northeast Atlantic is mainly caused by sulfate.

### 3.1.3  Contributions from the Mediterranean Sea and the Black Sea

The largest contributions to $PM_{2.5}$ concentrations from shipping in the Mediterranean and Black Sea region are calculated in and around the shipping lane from Gibraltar to the Suez Canal. High concentrations are also calculated in and around the Adriatic Sea and around some of the major ports like Marseille in France and Pireus in Greece. As in the NE Atlantic sulfur emissions from shipping are high, and the summer maximum in this sea area is mainly caused by sulfate.

### 3.1.4  Contributions from Rest of world shipping

Given the large distance to the European continent, contributions to European $PM_{2.5}$ levels from ROW shipping are small.

### 3.1.5  Country attributions

The source receptor relationships for shipping (total and from separate sea areas) are listed in Table 3 for selected countries. Here we also list the corresponding source receptor results as reported in the latest EMEP report (EMEP Status Report 1/2019, 2019). In general, the reported relationships and the results from the global model are in good agreement. Differences between the global and regional calculations are discussed in Section 5.

In Figure 4 the percentage contributions from all ships and from emissions in different sea areas to selected European countries are shown. The contributions are calculated from the scenarios listed in Section 2.3. We let the difference between the Base_2017 and the SR_All represent 100% of the anthropogenic contributions to $PM_{2.5}$. The contributions from the individual sea areas are stacked on top of each other. The stacked contributions are shown in parallel to the contributions from all ships (Base_2017 - SR_AllSh). Any difference in the length of the two bars can be interpreted as a measure of non-linearities in the calculations. Moderate deviations from linearity are in particular seen for the countries bordering the southern parts of the North Sea, caused by differences in ammonium nitrate formation between the model scenarios. The contributions from all ships are split into a black and a grey part where the first grey part represents the contributions with CAP2020 sulfur emissions and the black part the additional contributions when using 2017 emissions, i.e. prior to the implementation of CAP2020. The effects of the CAP2020 regulations are discussed in more detail in Section 4.

The figure clearly shows that the countries are most affected by nearby ship emissions, in particular in smaller countries close to major shipping lanes. Malta with about 50% of the anthropogenic contribution, Cyprus almost 20% and Greece almost 15%, in the Mediterranean Sea, and Denmark with about 15%, bordering both the North Sea and the Baltic Sea, are some of the countries most affected. Countries bordering only the Mediterranean Sea and the Black Sea are hardly impacted by other sea areas. A few countries are bordering more than one of the separate sea areas. This is exemplified by Norway (about 10%) and UK (about 10%) which are strongly impacted by ship

emissions in both the North Sea and the remaining Atlantic. Spain (about 15%) is impacted by both the remaining Atlantic and the Mediterranean Sea. France (about 8%) is a "tricolore" country affected by ship emissions in the North Sea, Mediterranean Sea and remaining Atlantic.

## 3.2 Ozone

Figure 2c shows the global concentration of $O_3$ and Figure 2d the contributions from global shipping. Globally the highest concentrations are calculated for the latitudinal band between 20 to 40 degrees north. The largest contributions from shipping are mainly calculated in and around the major ship tracks in south Asia, resulting from high $NO_x$ emissions in combination with favourable meteorological conditions for ozone production. In Europe there are similar favourable conditions in and around the Mediterranean Sea. Below we discuss how ship emissions from different sea areas affect European ozone levels split by season.

Net formation of ozone depend on the ratio between $NO_x$ and NMVOC. In regions with high $NO_x$ concentration ozone production is limited by the availability of NMVOC, and further enhancements of $NO_x$ will lead to increased ozone titration, and thus reductions of ozone, predominantly in the winter months. In summer additional NMVOC emissions from leisure boats may lead to an increase in ozone levels in such areas. In areas limited by the availability of $NO_x$ additional $NO_x$ will result in increased ozone production, predominantly in the summer months.

### 3.2.1 North Sea and Baltic Sea

In the North Sea and Baltic Sea regions (Figure 5a,b,c,d), ship emissions contribute to widespread ozone titration in all four seasons. The strongest titration effects are calculated in winter and the least in summer.

### 3.2.2 Northeast Atlantic

Although there is a net ozone loss throughout much of the year in the shipping lane from Gibraltar to the entrance of the English Channel, shipping contributes to higher ozone in most of the bordering countries all year with the exception of the UK, northern Scandinavia and coastal regions next to the shipping lanes. Net ozone production is in particular high in summer (Figure 5 e,f,g,h).

### 3.2.3 Mediterranean Sea and Black Sea

In the Mediterranean Sea and the Black Sea there is widespread ozone titration close to major shipping lanes and ports in winter (Figure 5i,j,k,l). However, in Spring ozone production starts to dominate, reaching a maximum in summer with contributions from shipping of more than 4ppb in the eastern Mediterranean sea and bordering land areas.

### 3.2.4   Rest of world shipping

Emissions from Rest Of World shipping affects all of Europe, but western and northern Europe more than southern and eastern Europe (Figure 5 m,n,o,p). The seasonal behaviour differs from the other sea areas with a summer minimum and a slight winter maximum. On an annual basis contributions are comparable, and in some regions higher, than contributions from the other sea areas. This is shown in more detail in the section about country attributions below.

### 3.2.5   Country attributions

For SOMO35[3] the source receptor relationships for shipping (total and from separate sea areas) are listed in Table 4 for selected countries. We also list the corresponding source receptor calculations as reported in the latest EMEP report (EMEP Status Report 1/2019, 2019) and these results are discussed in Section 5.

In Figure 6a,c the contributions from all ship emissions and from emissions in different sea areas to selected European countries are shown for annually averaged ozone in ppb following 15% reductions in ship emissions in the sea areas. The calculated effects of 15% reductions in all anthropogenic emissions are given as numbers in the figure. In Figure 6b,d the effects of ship emissions on SOMO35 are given as a percentage of the total anthropogenic contributions. Given the non-linear behaviour of the ozone chemistry, contributions from the separate sea areas are not stacked (as for $PM_{2.5}$ in Figure 3). The full length of the bars are split so that the first, darker part, of the bars represent the calculations with the ShipNOX parameterisation included as described in Section 2.3 and the second, brighter coloured part, the calculations without ShipNOX. The difference between the calculations with and without ShipNOX can be interpreted as a range for the effects of ship emissions on ozone levels. In Belgium, the Netherlands, and Malta the contributions from anthropogenic emissions, and also from ship emissions, to annually averaged ozone are negative as a result of ozone titration.

Contrary to what was shown for $PM_{2.5}$ there are significant contributions from ROW shipping in most countries. Ranging from about 5% to 8% for countries bordering the North Sea and the Baltic Sea. Jess for Mediterranean and Black Sea countries. For several countries in western and northern Europe, and in landlocked countries exemplified by Austria, as well as in Romania (partially bordering the Black Sea) (Figure 6), ROW shipping is the largest contributor to anthropogenic ozone levels both with regard to SOMO35 and annual average ozone. In the Mediterranean countries the by far largest contributions come from Mediterranean shipping with contributions up to 20% for Cyprus. In these countries the second largest contributions are from ROW shipping. In and around the southern part of the North Sea both land based and ship emissions of $NO_x$ are high, and as also shown in Figure 5a,b,c,d, ozone levels decrease as a result of North Sea and Baltic Sea shipping. For the overall effects of shipping this decrease is compensated by contributions from other sea areas. However, in

---

[3]SOMO35 is the indicator for health impacts recommended by WHO calculated as the daily maximum of 8-hour running ozone maximum over 35 ppb

Belgium, the Netherlands and also Malta in the Mediterranean Sea, the overall contributions of ship emissions from all sea areas give reductions in annually averaged ozone levels of the order of 0.1 to 0.2 ppb. In Belgium and the Netherlands we also calculate reductions in SOMO35 from shipping in the Netherlands by more than 10% of the contributions from all anthropogenic sources.

### 3.3   Depositions of sulfur and oxidised nitrogen

Figure 2e,g shows the total (wet and dry) depositions of oxidised nitrogen and sulfur centred around Europe. For oxidised nitrogen large depositions are calculated in north central Europe and in the Po Valley in Italy. For sulfur the largest calculated depositions are mainly calculated in eastern Europe.

Figure 7 shows the contributions from the separate sea areas to depositions of oxidised nitrogen and sulfur of antropogenic origin to selected countries. In addition, the source receptor relationships are
listed in Tables 5 and 6 for both global and regional model calculations. Depositions from shipping are largely confined to areas/countries near the sea, peaking close to major shipping routes. For most of the coastal countries the percentage contributions to depositions of oxidised nitrogen are more than 20%. Even for lager countries, as Germany, Poland , France, and Spain, the percentage contributions are 10% or more. Sulfur depositions from shipping are low in and around the North Sea and Baltic
Sea where sulfur emissions are low as a result of the SECA regulations. Even so, contributions from shipping are ranging from 3% to more than 10% for these countries. For other coastal countries contributions range from 10% to almost 70% for Malta.

## 4   Effects of CAP2020 on European $PM_{2.5}$ levels and on sulfur depositions

From January 1th 2020 the maximum allowed sulfur content in marine fuels was reduced to 0.5%
(CAP2020). Before CAP2020 the global average sulfur content outside SECAs was around 2.5% although a higher percentage sulfur content of 3.5% was allowed. The latest figures showed that the yearly average sulfur content of the residual fuel oils tested in 2017 was 2.54, see http://www.imo.org/ en/MediaCentre/HotTopics/GHG/Documents/2020%20sulphur%20limit%20FAQ%202019.pdf, last access 7 July 2020. Our calculations show that prior to CAP2020 the fraction of sulfate in $PM_{2.5}$ is
low in the North Sea and Baltic Sea, as well as in most of continental Europe and the British Isles as a result of the SECA regulations. However, in sea areas outside the SECA, and in land areas bordering these sea areas, sulfate is the major component in $PM_{2.5}$ origination from ship emissions (Figure 8a).

To give an estimate of the effects of CAP2020 on European $PM_{2.5}$ levels and the depositions of oxidised sulfur we have made calculations reducing sulfur emissions outside the North Sea and the
Baltic Sea SECAs by 80%, corresponding to a reduction from 2.5% to 0.5% in the sulfur content in the fuels. This is a crude estimate, as there are low emission ships operating outside the SECAs. On the other hand CAP2020 compliance may not reach 100%. Furthermore we have assumed 80% reductions in sulfur emissions also in low emissions zones far from European waters. But, as already

shown in Figure 3, emissions outside European waters (ROW shipping) has little or no effects on European $PM_{2.5}$ levels. Sofiev et al. (2018) estimated a 75% reduction of global sulfur emissions. As sulfur emissions are already below the CAP2020 levels in the SECAs, this is close to the reduction assumed in this study.

Figure 8b shows the calculated effects of CAP2020 on European $PM_{2.5}$ levels. Reductions in $PM_{2.5}$ ranging from 0.5 to more than $2\mu gm^{-3}$ are calculated in the major shipping routes in the Mediterranean and eastern Atlantic ocean, affecting also neighbouring land areas where ship emissions make up a significant percentage of the $PM_{2.5}$ concentrations. In Sofiev et al. (2018) they calculate similar reductions, ranging from 2 to $4\mu gm^{-3}$, in major shipping lanes, but the largest reductions are calculated for sea areas outside European waters. In European waters north of 62 degrees the sulfur fraction is also high, but here ship traffic is much lower and the effects on $PM_{2.5}$ well below $0.1\mu gm^{-3}$.

In Figure 4 the contributions to $PM_{2.5}$ from all ships to selected European countries are shown as a percentage of all anthropogenic contributions calculated with ship emissions before and after the implementation of CAP2020. In particular in the countries bordering the Mediterranean Sea, the percentage contributions to $PM_{2.5}$ relative to all anthropogenic emissions are reduced by about 50%. For countries bordering the North Sea and the Baltic Sea SECA, where sulfur emissions prior to CAP2020 are very low, the percentage reductions in the contributions to $PM_{2.5}$ are much smaller.

A similar pattern as $PM_{2.5}$ is seen in Figure 7 for oxidised sulfur depositions, with substantial reductions in depositions of anthropogenic origin in countries bordering sea areas that are not SECAs, such as the Mediterranean Sea and the Northeast Atlantic.

## 5  Differences between regional and global model calculations

The regional model calculations as reported in the annual EMEP reports (exemplified by the latest EMEP report, EMEP Status Report 1/2019 (2019)) are widely used for regulative purposes within the EU and for the LRTAP convention (Convention on Long-Range Transboundary Air Pollution, (http://www.unece.org/fileadmin//DAM/env/lrtap/welcome.html, last accessed 7 July 2020). The alternative global calculations presented here gives an indication of the robustness of the officially reported calculations.

In general the results from the global and the regional model calculations are in good agreement. Even so there are some systematic differences in the model results. We have tried to trace these to differences listed below in model input and model setup, and to what extent global and regional calculations could give qualitatively and quantitatively different results for the effects of ship emissions.

   1. As discussed in section 2.2, Land based emissions are not identical.

2. The ship emission sets used in the global and regional calculations have a common origin (see section 2.2). Even so, annual emission totals for the individual sea areas differ. In the global calculations ship emissions in the individual sea areas are in general higher.

3. In the global model we reduce the emissions by 15% for all species in the sea areas simultaneously, whereas in the regional calculations emissions of the individual species are reduced separately.

4. The resolution used in the global and regional model calculations differ.

5. In the regional calculations the boundary and initial conditions for all gaseous and aerosol species were given as 5-year monthly average concentrations, derived from EMEP MSC-W global runs.

Bullet points 3 and 4 were a compromise to keep the computational demand of the global calculations within reasonable limits. Below we discuss the effects this makes for different components in detail. We also make statements on the processes behind these difference, which is of relevance also beyond this study.

### 5.1 Differences in $PM_{2.5}$

For almost all countries bordering the Baltic Sea and North Sea, the effects of ship emissions on $PM_{2.5}$ are consistently higher in the global versus the regional calculations (see Table 3). In most cases this is because the ship emissions used in the global model are higher than in the regional model calculations (see Section 2.2). There are also some additional factors causing differences:

Most countries bordering the Baltic Sea and the North Sea are high emitters of ammonia. $SO_4$ (either emitted directly or oxidised from $SO_2$) can react with ammonia forming ammonium sulfate. Much of the emitted $NO_x$ will form $HNO_3$. Given ammonia in excess of $SO_4$, $HNO_3$ will react with ammonia forming ammonium nitrate. As shown in Table 1 emissions of in particular sulfur in the European Union (and subsequently in countries bordering these two sea areas) are higher in the global model calculations. In addition sulfur emissions are slightly higher in the remaining Northeast Atlantic. As a result, more sulphate is available for ammonium sulfate formation and thereby allowing less of the $HNO_3$ from shipping to form particulate ammonium nitrate. This explains the lower formation of $PM_{2.5}$ from shipping in the vicinity of regions of high ammonia emissions.

In several countries $PM_{2.5}$ levels from shipping are markedly higher in the global calculation, in particular in small countries such as Cyprus, and also in Portugal where the shipping lanes are very close to the shore. We believe this is caused by the lower resolution in the global calculations, which implies that grid boxes covering partially land and sea extend further inland, thus artificially extending the effect of ship emissions somewhat further into these countries' territories.

**5.2 Differences in nitrogen and sulfur deposition between global and regional model calculations**

Depositions of both oxidised nitrogen and sulfur are in general higher in the global model calculations as a result of higher emissions used in the global model. Above we argued that parts of the lower contributions from ships to $PM_{2.5}$ concentrations could be caused by less ammonia available for ammonium nitrate formation in the global calculations, resulting in a higher $HNO_3$ to ammonium nitrate ratio. As the dry deposition of $HNO_3$ is faster than for ammonium nitrate, more oxidised nitrogen (mainly ammonium nitrate, $HNO_3$, $NO_2$) is deposited in nearby countries where ammonia emissions are high.

In several countries both nitrogen an sulfur depositions are higher in the global model calculations than what can be explained by differences in emissions alone, in particular in small countries such as Malta and Cyprus, and in Portugal where the shipping lanes are very close to the shore. As for $PM_{2.5}$ concentrations, we believe this is caused by a lower resolution in the global calculations as grid boxes covering partially land and sea extend further inland.

**5.3 Differences in SOMO35**

In Table 4 the contributions from ship emissions to selected countries are listed, both for the global and regional model calculations. Given the large compensating contributions from ozone titration, mainly in winter, and ozone production, mainly in the summer months, SOMO35 calculated with the global and the regional model versions are remarkably similar. However, there are substantial differences, mainly confined to the very high $NO_x$ emitting regions bordering the North Sea.

In the global calculations there are substantial contributions from ROW shipping (see Table 4) that can not be attributed in the regional calculations, and in several countries ROW is the largest contributor (see section 3.2.5).

With the ShipNOX parameterization included in the global calculations the contributions to SOMO35 from the sea areas is reduced by about 50% (see Figure 6) and considerably lower than in the regional calculations. ShipNOX is not used in the regional calculations, but the largest effects of ignoring the ship plume chemistry should be in low $NO_x$ areas with large gradients between the plumes and ambient air most often found in pristine sea areas.

**6 Conclusions**

Emissions from shipping are large sources of air pollution and depositions of oxidised nitrogen and sulfur. In this study we have mainly restricted ourselves to the effects on European pollution levels, but the effects are global. In particular in coastal regions/countries, we attribute a large portion of the $PM_{2.5}$ of anthropogenic origin to emissions from shipping. For $PM_{2.5}$ we show that the largest contributions come from nearby waters. The calculations show that contributions from sulfur to

PM$_{2.5}$ are low from the North Sea and the Baltic sea where the strict SECA regulations apply. Prior to the implementation of the CAP2020 regulations between 50% and 80% of the the anthropogenic PM$_{2.5}$ mass in countries/regions not bordering the SECAs was from sulfate. Here sulfate levels peak in summer when the conversion rate of SO$_2$ to sulfate is at its highest. In the SECA sea areas nitrates (mainly ammonium nitrate) is the largest constituent in anthropogenic PM$_{2.5}$, peaking in Spring as a result of the large ammonia emissions in nearby land areas in this season. With additional sulfate and gas phase HNO$_3$ from ship emissions, more ammonium (ammonium nitrate and ammonium sulfate) is formed, contributing about 20% - 30% of the PM$_{2.5}$ dry mass from shipping in much of the European land areas. As a result, the combination of sulfur and NO$_x$ emissions from shipping further increase the PM$_{2.5}$ burden in and around regions with high ammonia emissions beyond what strictly speaking is originating directly from SO$_x$ and NO$_x$. Without ship emissions a larger portion of the ammonia would have been deposited to the surface and not contributing to the particle formation.

The very low fraction of sulfate in PM$_{2.5}$ in and around the North Sea and the Baltic Sea demonstrates the effectiveness of the SECA regulations in reducing the PM$_{2.5}$ burden from shipping here. A global sulfur cap was implemented from January 1th 2020. Assuming the fulfilment of the legislation, it is expected that this now has resulted in substantial reductions in the PM$_{2.5}$ burden globally. This has resulted in approximately 50% reductions in calculated PM$_{2.5}$ from shipping in European countries and regions not bordering the SECAs. In a similar study, using the SILAM model, Sofiev et al. (2018) calculated reductions in PM$_{2.5}$ levels in the busiest sea-lanes of 2 to 4 $\mu$gm$^{-3}$. This is similar to the total contribution from shipping shown in Figure 2 (100% versus 80% sulfur control).

In Karl et al. (2019) the SILAM model, along with the CMAQ model, was compared to the EMEP model focusing on the effects of ship emissions in the Baltic Sea in 2012, prior to the implementation of stricter SECA regulations in 2015. As noted in section 2.1 the CMAQ and the EMEP models had a slightly negative bias for PM$_{2.5}$, whereas the SILAM model had virtually no bias. Even so, the SILAM model calculated a slightly lower contribution from Baltic Sea shipping compared to the other two models. In this study, using year 2012 emissions, the average contribution of ships to PM$_{2.5}$ levels ranged between 4.15 and 6.5 % in the entire Baltic Sea region, and between 3.15 and 5.7 % in the coastal land areas. For year 2017 these contributions are considerably lower, as was also shown in Jonson et al. (2019).

In Chinese coastal regions the peak contributions to PM$_{2.5}$ concentrations in this study are lower than in the study by Lv et al. (2018). There are several possible explanations for this difference. Lv et al. (2018) used a finer model resolution ($36 \times 36$ km) than in the present study. A finer resolution is likely to result in somewhat higher peak concentrations. Stricter regulations, limiting the sulfur content in marine fuels to 0.5% in and around several Chinese ports, including the YRD (Yangtze River Delta), have been imposed between these two studies (2013 versus 2017), and are included in the ECCAD 2017 ship emission data. According to Lv et al. (2018) YRD is responsible for about 20% of the ship emissions in Chinese waters.

The net effects on surface ozone from ship emissions is a combination of ozone destruction, mainly in winter, and ozone production, mainly in summer.This is also the reason for the different behaviour of annual averaged ozone and the SOMO35 ozone metric. SOMO35 is hardly accumulated in winter when ozone titration events are most frequent as ozone levels in winter are regularly below the 35ppb threshold.

The lifetime of ozone in the atmosphere is considerably longer than for$PM_{2.5}$ ranging from hours to a few days in the boundary layer to weeks and even months in the free troposphere (TF HTAP, 2010). As a result ozone can be transported at intercontinental scales, explaining the large contributions from ROW shipping.

Global model calculation require substantially more computer power than regional calculations, and thus global scale source receptor calculations, even with a half a degree resolution, would not be possible with all countries and regions included in the regional scale calculations. The source receptor relationships derived from the global and regional calculations are similar. Where there are differences, these can largely be attributed to model setup and input data. Most of species levels, and the resulting surface depositions, highlighted in EMEP regional calculations are relatively short-lived. As a result the effects of emissions originating outside the regional model domain are small. Thus the additional benefits of global model calculations are small compared to the improvements in accuracy that can be achieved with finer resolution on a smaller model domain. For ozone, enhancing the resolution improves the representation of localised variations in $NO_x$ to NMVOC ratios, explaining the differences in particular in the high $NO_x$ emitting countries and regions bordering the North Sea. On the other hand, with global scale calculations the contributions to ozone from all global sources can be included. For several countries/regions we show that for ozone, contributions from ROW shipping are comparable, and in some regions higher than the contributions from sea areas close to Europe. In the regional model source receptor calculations bic (boundary and initial concentrations) only account for ozone 'produced within the regional model domain from $NO_x$ (emissions of NMVOC from shipping are very small) transported from outside the regional model domain.

The dispersion and chemistry in the shipping plumes represent an uncertainty in the calculations. Calculations including the "ShipNOX" parameterisation short-circuit the $NO_x$ chemistry so that only half of the emitted $NO_x$ enters the ozone cycle, and as a result, the effect of shipping on ozone is also reduced by about 50%. Calculations with and without the "ShipNOX" parameterisation give an upper and lower range for the effects of shipping on ozone. The largest effects of ship plume chemistry are likely to occur where the gradients between ship plume and ambient air $NO_x$ concentrations are large. Such conditions are less common in waters close to Europe. In their plume calculations, Vinken et al. (2011) reported that almost all the ozone was depleted in the first stages of the plume. In Karl et al. (2019) the EMEP model, with its coarser grid resolution, calculated less ozone titration in the shipping lanes. However, downwind of the shipping plumes ozone is regenerated. As a result, the impact of ship emissions on ozone in nearby land areas was comparable for the EMEP and CMAQ

models, but lower for the SILAM model. These results partially corroborate the chemistry in plumes
outlined by Vinken et al. (2011), but also demonstrate the regeneration of ozone downwind of the
ship plumes.

*Acknowledgements.* This work has been partially funded by EMEP under UNECE. Computer time for EMEP
model runs was supported by the Research Council of Norway through the NOTUR project EMEP (NN2890K)
for CPU, and NorStore project European Monitoring and Evaluation Programme (NS9005K) for storage of data.
The National Center for Atmospheric Research is funded by the National Science foundation. The ship emission
data have been downloaded from the ECCAD database https://eccad.aeris-data.fr/.

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

**Table 1.** Ship emissions from FMI in European sub sea areas. Sulfur emissions are given as $SO_2$. PPM emissions are sub-divided into Ash, EC and OC, all assumed emitted as $PM_{2.5}$. Total EU emissions used in global and regional calculations are also listed. 5% of these $SO_2$ are assumed to be emitted as $SO_4$.

| | **Sulfur** | | **NOx** | **CO** | **PPM** | | | **NMVOC** |
|---|---|---|---|---|---|---|---|---|
| | Gg $SO_2$ | | Gg $NO_2$ | Gg CO | see caption | | | Gg as C |
| | $SO_2$ | $SO_4$ | | | Ash | EC | OC | |
| Global | 9408 | 559 | 19670 | 1360 | 91 | 124 | 309 | 150 |
| Mediterranean Sea | 680 | 40 | 1340 | 92 | 6.4 | 8.7 | 22 | 11 |
| Black Sea | 66 | 3.8 | 158 | 12 | 0.8 | 1.1 | 2.7 | 1.3 |
| Baltic Sea | 9.9 | 0.7 | 313 | 21 | 1.5 | 2.0 | 4.9 | 2.6 |
| North Sea | 27 | 1.6 | 684 | 52 | 3.4 | 4.6 | 11.8 | 5.8 |
| Remaining Atl. | 456 | 27 | 836 | 60 | 4.0 | 5.4 | 13.5 | 6.5 |
| | | | European Union emissions | | | | | |
| EU Global 2017 | 2621 | | 7723 | 18227 | 1490 | | | 6245 |
| EU EMEP 2017 | 2274 | | 7537 | 25737 | 1303 | | | 7014 |

**Table 2.** Overview of model scenarios used. Separate model spin-up was only performed for base model run(s) and for model runs with globally perturbed emissions. For SR model runs perturbing limited areas we use the same spin-up as for the Base runs. CAP2020 emissions are estimated by scaling the emissions outside the North Sea and Baltic Sea SECAs from an assumed pre-CAP2020 global average sulfur content of 2.5% to 0.5%. Additional information about the model scenarios is given in section 2.3.

| Scenario | Description | spin-up |
|---|---|---|
| | Scenarios without ShipNOX | |
| Base_2017 | 2017 emissions unperturbed | 5 months |
| SR_AllAnt | All anthropogenic emissions reduced 15% | 5 months |
| SR_AllSh | All ship emissions reduced 15% | 5 months |
| SR_BALNOS | North Sea and Baltic Sea emissions reduced 15% | as Base_2017 |
| SR_MEDBLS | Mediterranean and Black Sea emissions reduced 15% | as Base_2017 |
| SR_ATL | Remaining NE Atlantic emissions reduced by 15% S | as Base_2017 |
| SR_ROW | Rest Of World ship emissions reduced 15% | 5 months |
| | Scenarios with CAP2020 | |
| CAP2020_Base | 2017 emissions unperturbed | 5 months |
| CAP2020_SR_AllAnt | All anthropogenic emissions reduced 15% | 5 months |
| CAP2020_SR_AllSh | All ship emissions reduced 15% | 5 months |
| | Scenarios with ShipNOX | |
| SHN_Base_2017 | 2017 emissions unperturbed | 5 months |
| SHN_SR_AllAnt | All anthropogenic emissions reduced 15% | 5 months |
| SHN_SR_AllSh | All ship emissions reduced 15% | 5 months |
| SHN_SR_BALNOS | North Sea and Baltic Sea emissions reduced 15% | as SHN_Base_2017 |
| SHN_SR_MEDBLS | Mediterranean and Black Sea emissions reduced 15% | as SHN_Base_2017 |
| SHN_SR_ATL | Remaining NE Atlantic emissions reduced by 15% S | as SHN_Base_2017 |
| SHN_SR_ROW | Rest Of World ship emissions rduced 15% | 5 months |

**Table 3.** Source receptor relationships for PM$_{2.5}$ from shipping. GL17 and GL20 calculated by the global model with 2017 and CAP2020 ship emissions respectively. The scenario calculations are made reducing the ship emissions for all species by 15%. "EMEP" is the source receptor calculations for 2017 from the latest EMEP report (EMEP Status Report 1/2019, 2019) appendix B. The EMEP source receptor reporting are based on separate calculations of individual species from all European countries and sea areas. **Glob** is the contribution from all global shipping, **NOS + BAS** from the North Sea and Baltic Sea combined, **MED + BLS** the Mediterranean Sea and Black Sea combined and **ATL** is the Northeast Atlantic. ROW includes all ship emissions outside the individual sea areas listed. For the "EMEP" reporting boundary and initial contributions are listed. Units: ng/m$^3$ per 15% emission reduction.

| Country | **Glob** | | **NOS + BAS** | | **MED + BLS** | | **ATL** | | **ROW** |
|---|---|---|---|---|---|---|---|---|---|
| | GL17 | GL20 | GL17 | EMEP | GL17 | EMEP | GL17 | EMEP | GL17 |
| Countries bordering the Baltic Sea | | | | | | | | | |
| Estonia | 22 | 21 | 20 | 22 | 0 | 0 | 2 | 1 | 1 |
| Latvia | 22 | 21 | 19 | 22 | 0 | 0 | 2 | 2 | 1 |
| Lithuania | 26 | 25 | 22 | 26 | 1 | 1 | 2 | 1 | 1 |
| Finland | 8 | 7 | 5 | 7 | 0 | 0 | 2 | 2 | 0 |
| Denmark | 110 | 107 | 99 | 112 | 1 | 0 | 8 | 6 | 3 |
| Sweden | 16 | 14 | 13 | 15 | 0 | 0 | 3 | 3 | 0 |
| Poland | 30 | 28 | 22 | 22 | 2 | 2 | 3 | 3 | 4 |
| Countries bordering the North Sea | | | | | | | | | |
| Belgium | 108 | 99 | 74 | 82 | 4 | 3 | 21 | 20 | 11 |
| Germany | 69 | 64 | 51 | 52 | 3 | 2 | 8 | 5 | 6 |
| Netherlands | 163 | 154 | 128 | 140 | 3 | 2 | 22 | 22 | 11 |
| Norway | 8 | 4 | 2 | 5 | 0 | 0 | 5 | 4 | 0 |
| GB | 68 | 52 | 28 | 34 | 1 | 1 | 35 | 33 | 3 |
| Countries bordering the North Atlantic | | | | | | | | | |
| Ireland | 42 | 29 | 12 | 11 | 0 | 0 | 28 | 28 | 2 |
| Portugal | 92 | 38 | 1 | 1 | 19 | 8 | 70 | 34 | 2 |
| Iceland | 5 | 2 | 1 | 1 | 0 | 0 | 4 | 3 | 0 |
| Countries bordering the Mediterranean and Black Sea | | | | | | | | | |
| Spain | 92 | 42 | 2 | 2 | 63 | 57 | 25 | 23 | 2 |
| France | 77 | 54 | 29 | 31 | 20 | 17 | 24 | 23 | 4 |
| Greece | 90 | 36 | 1 | 0 | 87 | 73 | 1 | 1 | 2 |
| Malta | 330 | 126 | 1 | 1 | 324 | 347 | 2 | 2 | 2 |
| Italy | 136 | 78 | 3 | 2 | 126 | 102 | 3 | 2 | 4 |
| Cyprus | 177 | 75 | 0 | 0 | 173 | 120 | 1 | 0 | 2 |
| Bulgaria | 21 | 11 | 1 | 1 | 18 | 21 | 1 | 1 | 1 |
| Romania | 17 | 11 | 3 | 3 | 11 | 12 | 2 | 1 | 1 |

**Table 4.** Source receptor relationships for SOMO35 from shipping as calculated by the global model, GL17, without SHIPNOX, see section 2, and as reported in EMEP Status Report 1/2019 (2019) appendix B. All the calculations are made with 2017 emissions and meteorological data. **Glob** is the contribution from all global shipping, **NOS + BAS** from the North Sea and Baltic Sea combined, **MED + BLS** the Mediterranean Sea and Black Sea combined and **ATL** is the Northeast Atlantic. ROW includes the effects from all ship emissions outside the above listed individual sea areas. BIC is regional Boundary and Initial Concentrations. Units: ppb.days per 15% emission reduction.

| | **Glob** | **NOS + BAS** | | **MED + BLS** | | **ATL** | | **ROW** | |
|---|---|---|---|---|---|---|---|---|---|
| Country | GL17 | GL17 | EMEP | GL17 | EMEP | GL17 | EMEP | GL17 | BIC |
| Countries bordering the Baltic Sea | | | | | | | | | |
| Estonia | 20 | 8 | 10 | 1 | 0 | 4 | 3 | 8 | 13 |
| Latvia | 22 | 9 | 10 | 1 | 0 | 4 | 3 | 8 | 14 |
| Lithuania | 22 | 8 | 9 | 1 | 1 | 5 | 4 | 8 | 15 |
| Finland | 15 | 3 | 4 | 1 | 0 | 3 | 3 | 7 | 10 |
| Denmark | 10 | -9 | -3 | 1 | 0 | 8 | 7 | 10 | 20 |
| Sweden | 18 | 3 | 6 | 1 | 0 | 5 | 4 | 9 | 14 |
| Poland | 19 | 5 | 5 | 1 | 1 | 5 | 4 | 8 | 18 |
| Countries bordering the North Sea | | | | | | | | | |
| Belgium | 1 | -15 | -10 | 1 | 1 | 6 | 7 | 8 | 20 |
| Germany | 14 | -33 | -2 | 1 | 1 | 6 | 6 | 9 | 21 |
| Netherlands | -12 | -26 | -18 | 1 | 0 | 6 | 6 | 7 | 18 |
| Norway | 23 | 4 | 5 | 1 | 0 | 7 | 5 | 12 | 17 |
| GB | 16 | -5 | -2 | 1 | 0 | 7 | 8 | 12 | 19 |
| Countries bordering the North Atlantic | | | | | | | | | |
| Ireland | 24 | -1 | 0 | 1 | 0 | 9 | 10 | 14 | 19 |
| Portugal | 47 | 1 | 0 | 3 | 5 | 25 | 28 | 17 | 41 |
| Iceland | 29 | 3 | 3 | 1 | 0 | 9 | 6 | 16 | 19 |
| Countries bordering the Mediterranean and Black Sea | | | | | | | | | |
| Spain | 37 | 1 | 0 | 7 | 13 | 13 | 14 | 16 | 39 |
| France | 26 | -0 | 0 | 5 | 7 | 10 | 10 | 12 | 25 |
| Italy | 43 | 3 | 1 | 24 | 33 | 5 | 4 | 10 | 25 |
| Greece | 46 | 3 | 2 | 30 | 35 | 3 | 2 | 10 | 26 |
| Malta | 53 | 3 | 1 | 31 | 22 | 6 | 4 | 13 | 25 |
| Cyprus | 115 | 2 | 0 | 100 | 75 | 2 | 1 | 11 | 27 |
| Bulgaria | 25 | 3 | 2 | 9 | 10 | 3 | 2 | 10 | 24 |
| Romania | 22 | 4 | 2 | 5 | 6 | 3 | 2 | 9 | 22 |
| Landlocked countries | | | | | | | | | |
| Austria | 24 | 3 | 2 | 4 | 5 | 5 | 4 | 11 | 23 |
| Switzerland | 26 | 2 | 1 | 4 | 6 | 6 | 5 | 13 | 26 |
| Czechia | 21 | 3 | 2 | 2 | 2 | 6 | 5 | 10 | 22 |

**Table 5.** Source receptor relationships for depositions of Dep of ox.N from shipping as calculated by the global model and as reported for year 2017. **Glob** is the contribution from all global shipping, **NOS + BAS** from the North Sea and Baltic Sea combined, **MED + BLS** the Mediterranean Sea and Black Sea combined and **ATL** is the Northeast Atlantic. GL17 are from the global model calculations, and EMEP are from EMEP Status Report 1/2019 (2019) appendix B. Units: 100 Mg of N per 15% emission reduction multiplied by 100/15.

| Country | Glob GL17 | NOS + BAS GL17 | NOS + BAS EMEP | MED + BLS GL17 | MED + BLS EMEP | ATL GL17 | ATL EMEP | ROW GL17 | ROW BIC |
|---|---|---|---|---|---|---|---|---|---|
| *Countries bordering the Baltic Sea* | | | | | | | | | |
| Estonia | 123 | 27 | 26 | 0 | 0 | 4 | 1 | 1 | 0 |
| Latvia | 39 | 36 | 34 | 1 | 1 | 2 | 1 | 1 | 1 |
| Lithuania | 37 | 34 | 30 | 1 | 1 | 2 | 1 | 1 | 1 |
| Finland | 99 | 88 | 84 | 1 | 1 | 8 | 6 | 1 | 10 |
| Denmark | 55 | 51 | 45 | 0 | 0 | 3 | 3 | 1 | 2 |
| Sweden | 201 | 183 | 163 | 1 | 1 | 15 | 13 | 2 | 16 |
| Poland | 166 | 143 | 126 | 7 | 6 | 10 | 8 | 6 | 11 |
| *Countries bordering the North Sea* | | | | | | | | | |
| Belgium | 38 | 31 | 25 | 1 | 1 | 5 | 4 | 2 | 4 |
| Germany | 286 | 238 | 197 | 9 | 8 | 27 | 23 | 13 | 29 |
| Netherlands | 72 | 62 | 48 | 1 | 1 | 7 | 6 | 2 | 6 |
| Norway | 116 | 88 | 86 | 1 | 1 | 26 | 22 | 2 | 22 |
| GB | 161 | 88 | 78 | 2 | 2 | 66 | 58 | 7 | 28 |
| *Countries bordering the North Atlantic* | | | | | | | | | |
| Ireland | 22 | 6 | 5 | 1 | 0 | 14 | 13 | 2 | 9 |
| Portugal | 51 | 1 | 1 | 9 | 8 | 39 | 34 | 2 | 10 |
| Iceland | 8 | 3 | 2 | 0 | 0 | 4 4 6 | 1 | 12 | |
| *Countries bordering the Mediterranean and Black Sea* | | | | | | | | | |
| Spain | 232 | 6 | 5 | 144 | 116 | 77 | 67 | 6 | 46 |
| France | 306 | 124 | 110 | 81 | 74 | 90 | 80 | 11 | 43 |
| Italy | 227 | 9 | 6 | 207 | 176 | 7 | 6 | 3 | 22 |
| Greece | 89 | 2 | 1 | 84 | 70 | 1 | 1 | 1 | 9 |
| Bulgaria | 32 | 4 | 3 | 27 | 23 | 1 | 1 | 1 | 5 |
| Romania | 46 | 13 | 10 | 28 | 25 | 2 | 1 | 1 | 8 |
| *Landlocked countries* | | | | | | | | | |
| Austria | 23 | 12 | 9 | 7 | 6 | 2 | 2 | 1 | 3 |
| Switzerland | 11 | 5 | 4 | 4 | 4 | 2 | 1 | 1 | 2 |
| Czech Rep | 29 | 22 | 17 | 3 | 2 | 2 | 2 | 2 | 3 |

**Table 6.** Source receptor relationships for depositions of Dep of ox.S from shipping as calculated by the global model and as reported for year 2017. **Glob** is the contribution from all global shipping, **NOS + BAS** from the North Sea and Baltic Sea combined, **MED + BLS** the Mediterranean Sea and Black Sea combined and **ATL** is the Northeast Atlantic. GL17 are from the global model calculations, and EMEP are from EMEP Status Report 1/2019 (2019) appendix B. Units: 100 Mg of S per 15% emission reduction multiplied by 100/15. Units: ppb.days per 15% emission reduction.

| Country | **Glob** | | **NOS + BAS** | | **MED + BLS** | | **ATL** | | **ROW** | |
|---|---|---|---|---|---|---|---|---|---|---|
| | GL17 | GL20 | GL17 | EMEP | GL17 | EMEP | GL17 | EMEP | GL17 | BIC |
| *Countries bordering the Baltic Sea* | | | | | | | | | | |
| Estonia | 2 | 1 | 1 | 1 | 0 | 0 | 0 | 0 | 0 | 3 |
| Latvia | 3 | 2 | 1 | 1 | 1 | 1 | 1 | 0 | 0 | 6 |
| Lithuania | 2 | 1 | 1 | 1 | 1 | 1 | 1 | 1 | 0 | 6 |
| Finland | 8 | 4 | 3 | 3 | 1 | 1 | 3 | 2 | 0 | 21 |
| Denmark | 5 | 4 | 3 | 2 | 0 | 0 | 2 | 1 | 0 | 6 |
| Sweden | 16 | 10 | 8 | 7 | 1 | 1 | 6 | 5 | 0 | 33 |
| Poland | 10 | 3 | 2 | 2 | 4 | 4 | 4 | 4 | 0 | 13 |
| *Countries bordering the North Sea* | | | | | | | | | | |
| Belgium | 6 | 4 | 3 | 2 | 0 | 0 | 2 | 2 | 0 | 4 |
| Germany | 39 | 23 | 19 | 9 | 6 | 6 | 12 | 10 | 1 | 50 |
| Netherlands | 15 | 12 | 12 | 5 | 0 | 0 | 3 | 2 | 0 | 5 |
| Norway | 25 | 9 | 5 | 6 | 1 | 0 | 19 | 15 | 0 | 45 |
| GB | 54 | 15 | 6 | 5 | 2 | 2 | 46 | 36 | 1 | 41 |
| *Countries bordering the North Atlantic* | | | | | | | | | | |
| Ireland | 12 | 3 | 0 | 1 | 1 | 0 | 14 | 13 | 2 | 9 |
| Portugal | 31 | 6 | 0 | 0 | 5 | 4 | 25 | 20 | 1 | 13 |
| Iceland | 8 | 1 | 0 | 0 | 0 | 0 | 3 | 3 | 0 | 22 |
| *Countries bordering the Mediterranean Sea* | | | | | | | | | | |
| Spain | 154 | 32 | 0 | 0 | 101 | 68 | 52 | 40 | 1 | 63 |
| France | 119 | 30 | 8 | 8 | 57 | 47 | 52 | 43 | 1 | 75 |
| Italy | 145 | 29 | 0 | 0 | 139 | 105 | 5 | 3 | 1 | 39 |
| Greece | 62 | 13 | 0 | 0 | 61 | 43 | 1 | 0 | 0 | 16 |
| Bulgaria | 17 | 3 | 0 | 0 | 16 | 13 | 1 | 0 | 0 | 16 |
| Romania | 21 | 4 | 0 | 0 | 19 | 16 | 1 | 1 | 0 | 23 |
| *Landlocked countries* | | | | | | | | | | |
| Austria | 6 | 2 | 1 | 0 | 5 | 4 | 1 | 1 | 0 | 11 |
| Switzerland | 4 | 1 | 0 | 0 | 3 | 3 | 1 | 1 | 0 | 7 |
| Czech Rep | 3 | 1 | 0 | 0 | 2 | 2 | 1 | 1 | 0 | 10 |

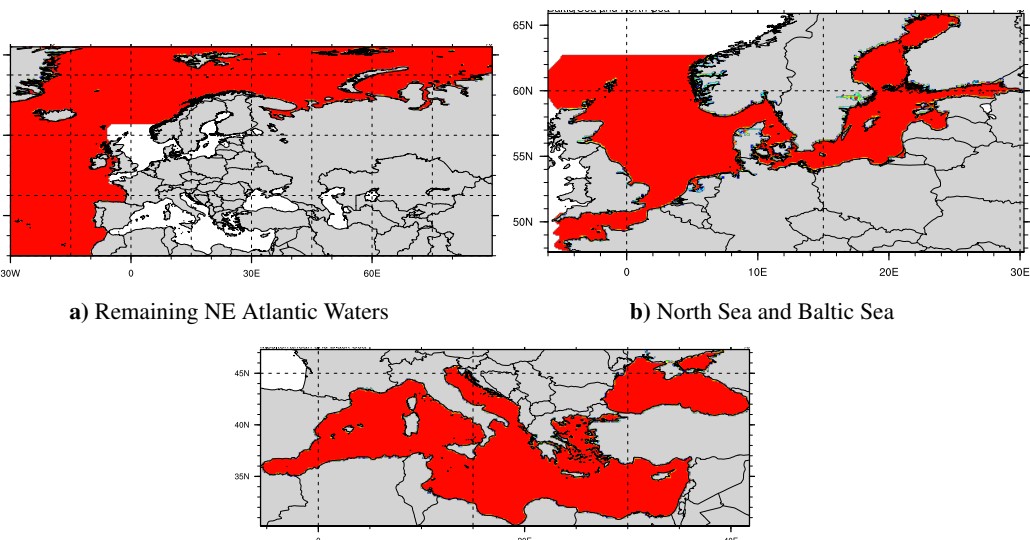

**a)** Remaining NE Atlantic Waters       **b)** North Sea and Baltic Sea

**Figure 1.** The individual sea areas marked in red. Shipping emissions in all other sea areas classified as ROW (Rest Of World) shipping.

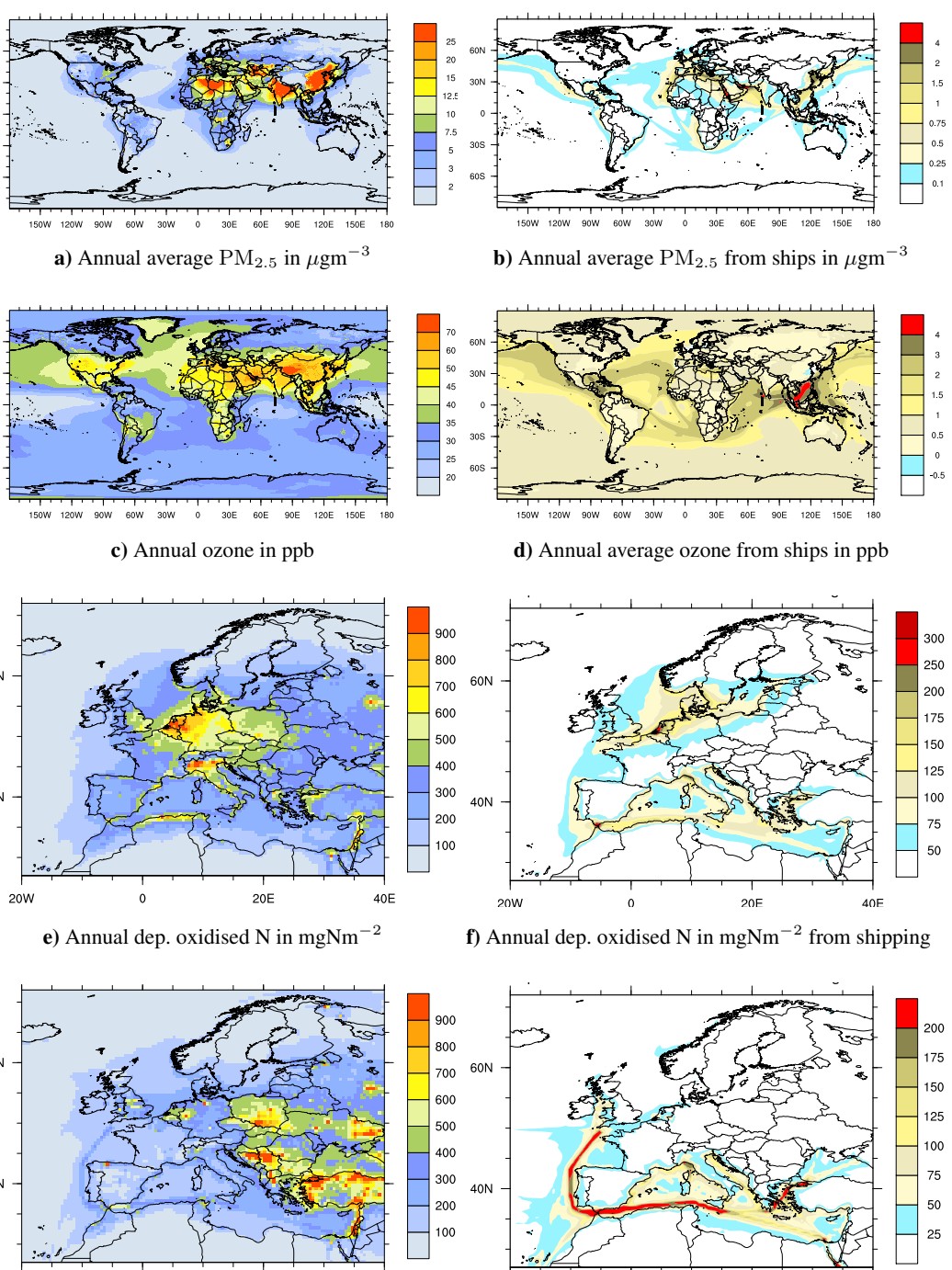

**Figure 2.** Right: Annually averaged global concentrations of $PM_{2.5}$ a) and $O_3$ c). Depositions of oxidised nitrogen e) and sulfur g). Left: Contributions from global shipping to $PM_{2.5}$ b) and $O_3$ d) and to depositions of oxidised nitrogen f) and sulfur h). The contributions from shipping have been multiplied by 100/15.

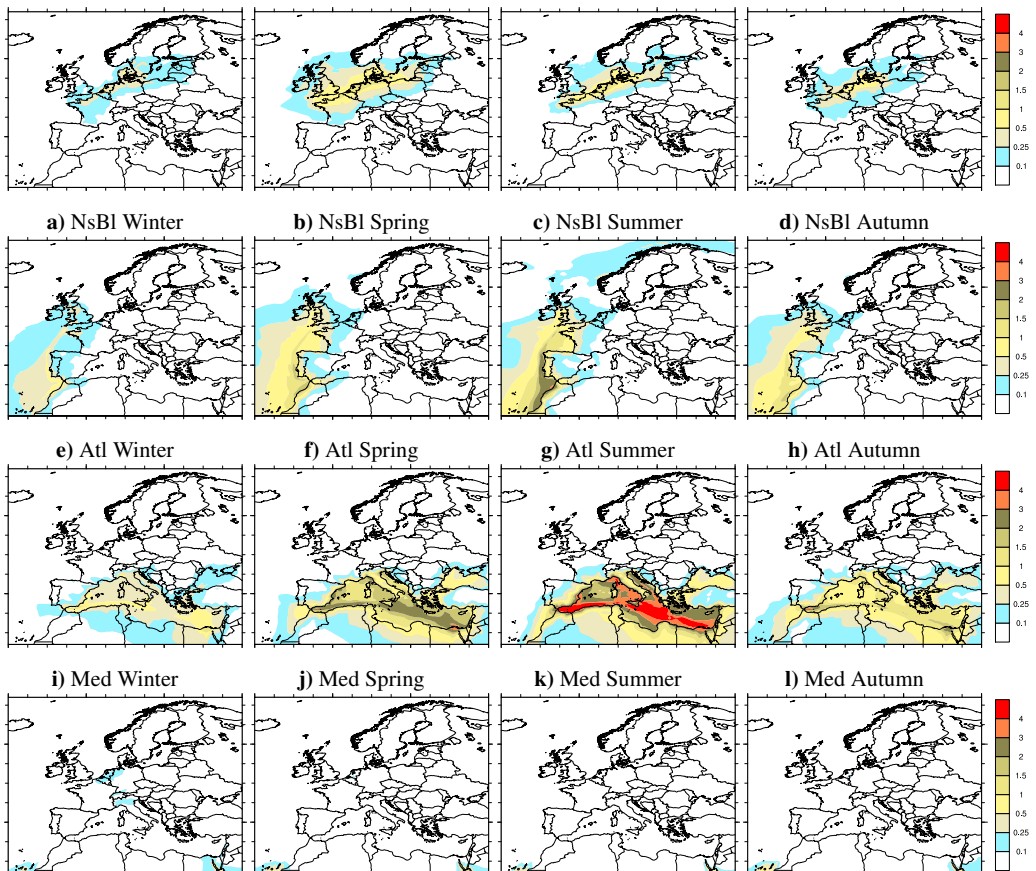

**Figure 3.** Seasonal contributions to European $PM_{2.5}$ levels (in $\mu gm^{-3}$) from 15% perturbations of the emissions in separate sea areas defined in section 2.3. Winter defined as December–January, Spring: March–May, Summer:June–August, Autumn: September–November. The contributions from shipping have been multiplied by 100/15.

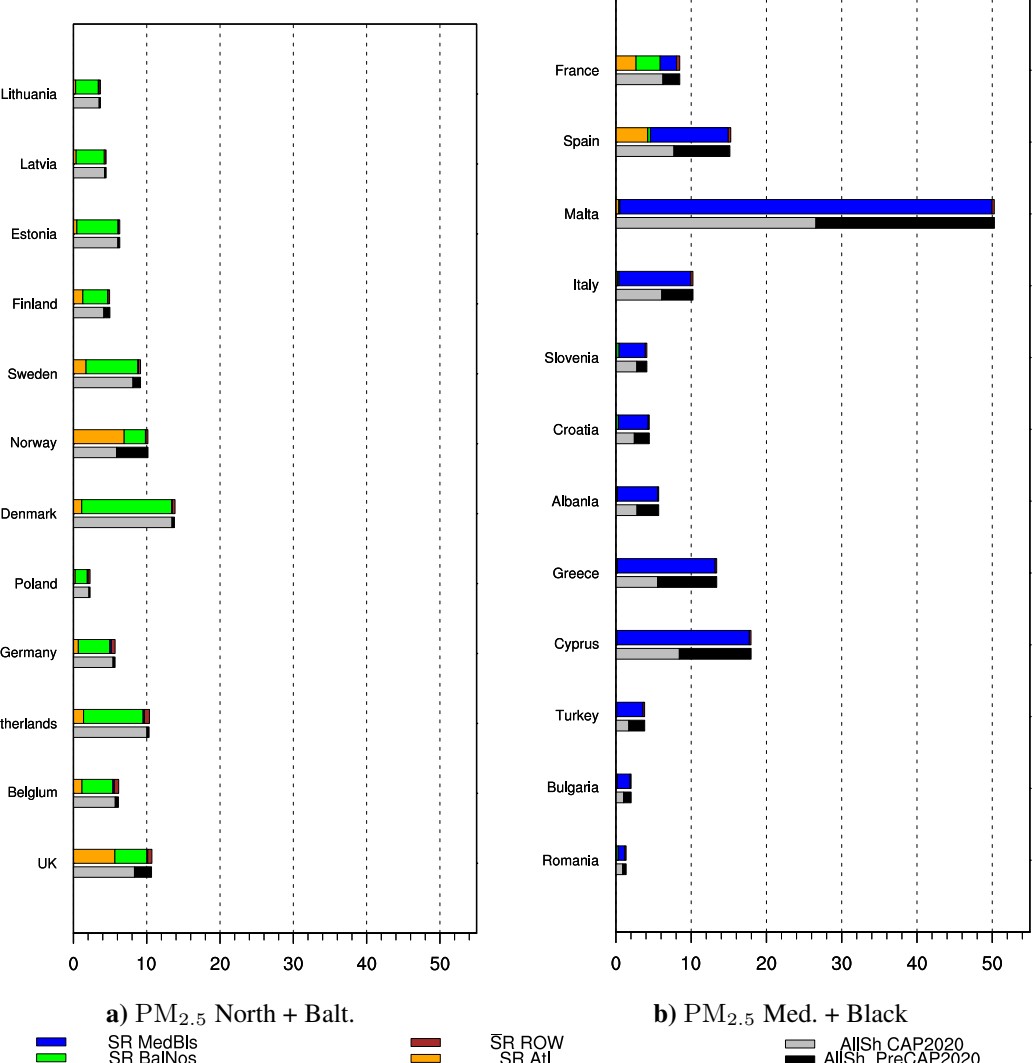

**Figure 4.** Percentage contributions from shipping to annually averaged $PM_{2.5}$ to countries bordering the North Sea and the Baltic Sea (left) and the Mediterranean Sea and the Black Sea (right) relative to contributions from all global anthropogenic emissions. Contributions are shown both for all ships and separated by sea area. For each country the contributions from the individual sea areas are added in the upper bar and the contributions from all ship emissions calculated as the difference between the Base and SR_AllShips scenarios are shown as black + grey bar below. The Base - SR_AllShips bars are split in a black and grey part where the first grey part represents the contributions after CAP2020 and black + grey the contributions prior to CAP2020. Differences in length between All ships (Black + grey) and the added contributions from the separate sea areas is an indication of non-linear effects.

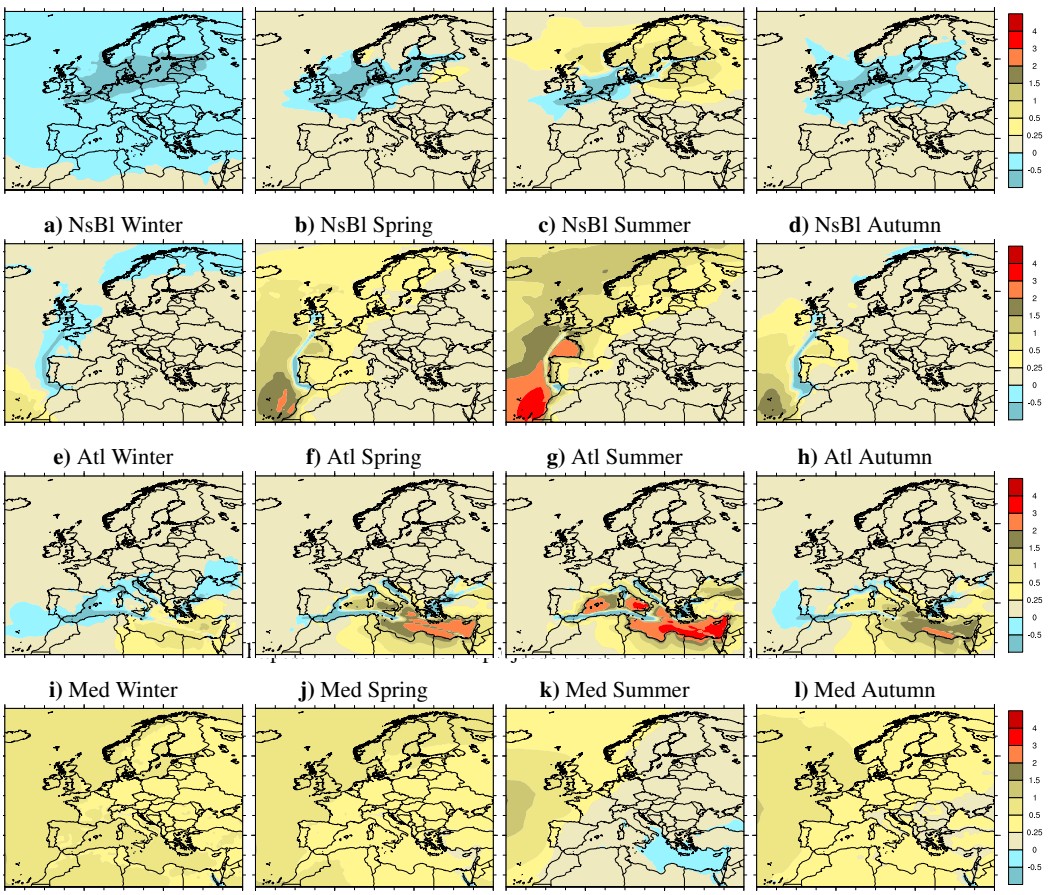

**Figure 5.** Seasonal contributions to European ozone levels (in ppb) from 15% perturbations of the emissions in separate sea areas defined in section 2.3. Winter defined as December–January, Spring: March–May, Summer:June–August, Autumn: September–November. The contributions from shipping have been multiplied by 100/15.

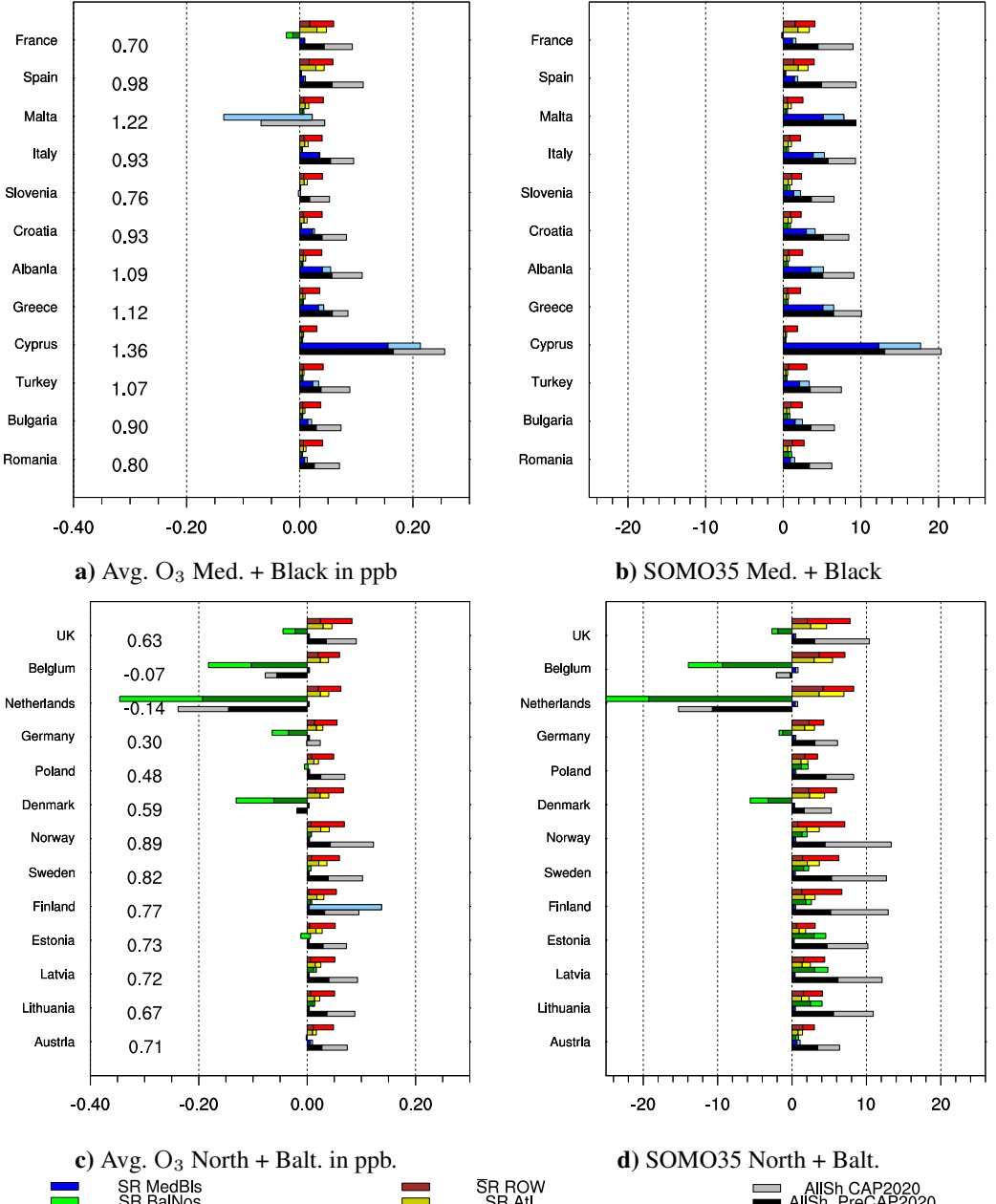

**Figure 6.** Contributions from shipping in ppb to annually averaged ozone from 15% reductions in ship emissions (left). Numbers to the right of the country names are the effects of the 15% reductions of all antropogenic emissions calculated as Base_2017 – SR_AllSh. Right, percentage contributions to SOMO35 relative to contributions from all global anthropogenic emissions. Contributions are shown for all ships and separated by sea area. The length of the bars are split so that the darker parts of the bars represent calculations assuming SHIPNOX (see section 2) and the full length without SHIPNOX. Note that for Malta the smaller perturbation in $NO_x$ from Mediterranean shipping with SHIPNOX results in a small ppb increase in calculated ozone, whereas the larger perturbation without SHIPNOX results in a decrease.

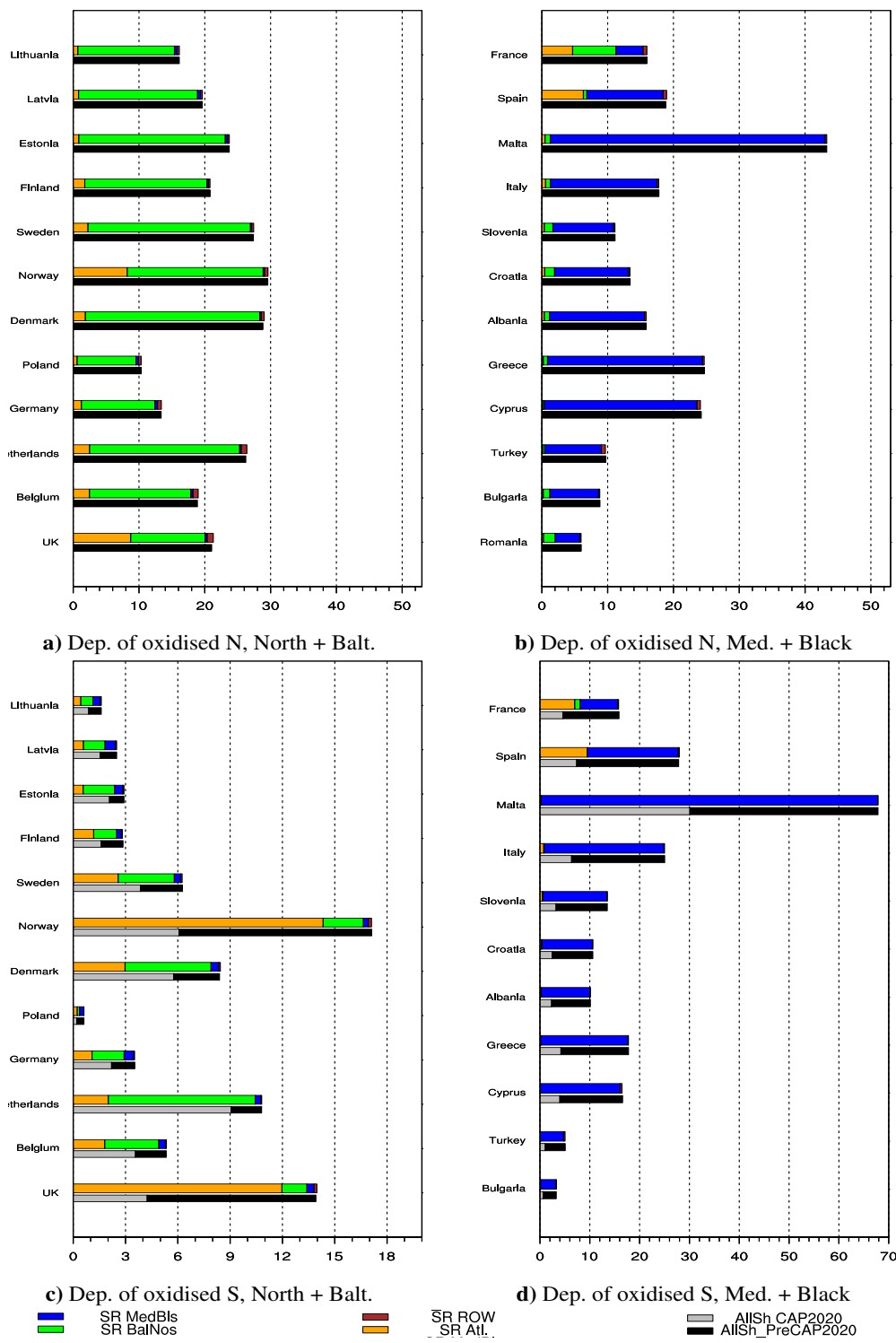

**a)** Dep. of oxidised N, North + Balt.

**b)** Dep. of oxidised N, Med. + Black

**c)** Dep. of oxidised S, North + Balt.

**d)** Dep. of oxidised S, Med. + Black

SR MedBls · SR ROW · AllSh_CAP2020
SR BalNos · SR Atl. · AllSh_PreCAP2020

**Figure 7.** Percentage contributions from shipping to annually averaged depositions of oxidised nitrogen (top) and sulfur (bottom) relative to contributions from all global anthropogenic emissions. Contributions are shown for all ships and separated by sea area. see also caption in Figure 4.

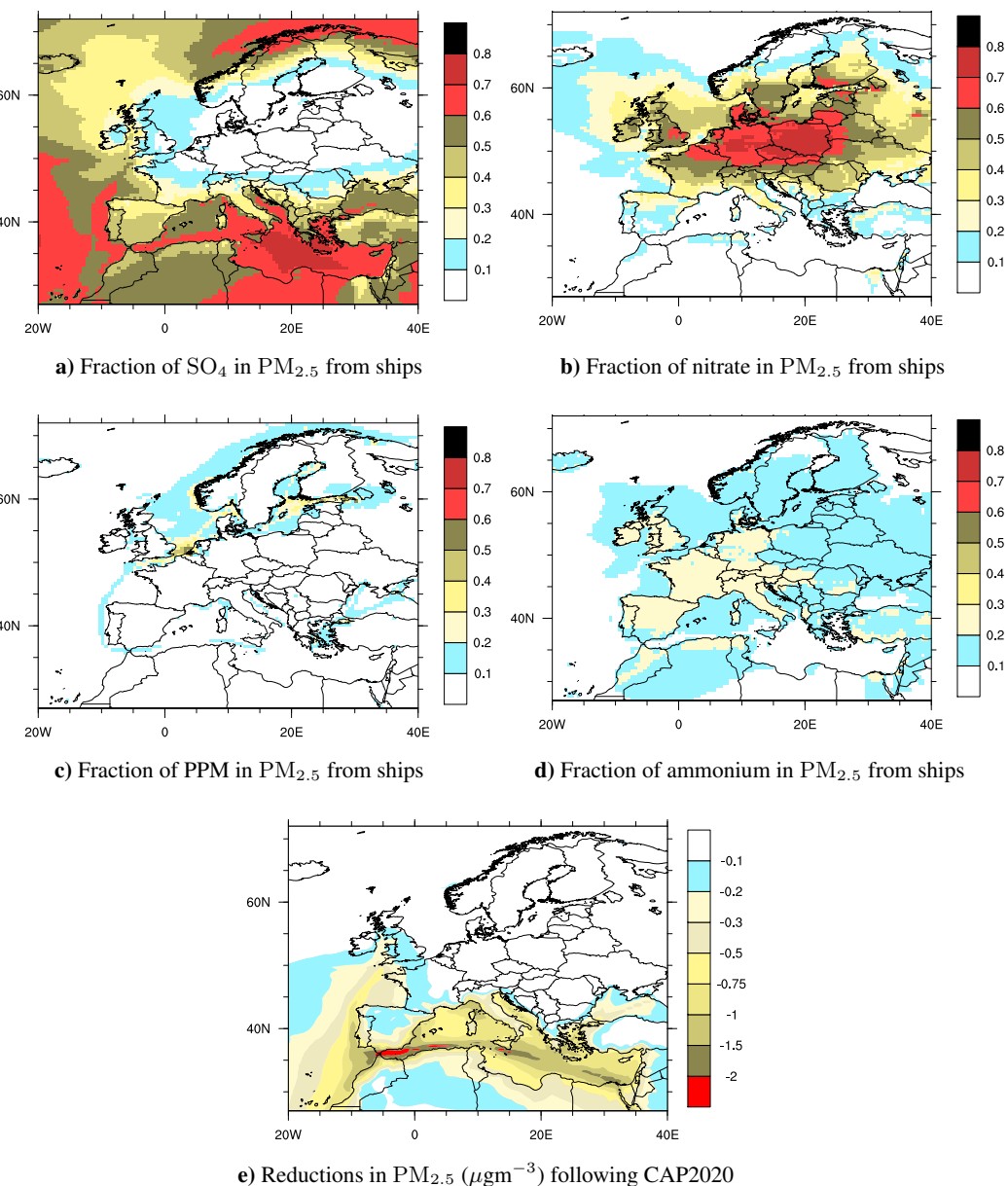

**a)** Fraction of $SO_4$ in $PM_{2.5}$ from ships

**b)** Fraction of nitrate in $PM_{2.5}$ from ships

**c)** Fraction of PPM in $PM_{2.5}$ from ships

**d)** Fraction of ammonium in $PM_{2.5}$ from ships

**e)** Reductions in $PM_{2.5}$ ($\mu gm^{-3}$) following CAP2020

**Figure 8.** Fraction of a) $SO_4$, b) nitrate, c) PPM and d) ammonia in $PM_{2.5}$ in European waters from shipping a). PPM are Ash, EC and OC. e) reductions in $PM_{2.5}$ ($\mu gm^{-3}$) following the CAP2020 regulations.