# Peer review of "Effects of global ship emissions on European air pollution levels"

_Atmospheric Chemistry and Physics, 2020_

## Referee Comment (RC1) · Mingxi Yang (Referee) · 13 May 2020

This is a very detailed assessment of the impact of ship emissions (both from nearby and far away locations) on air pollution levels in Europe, focusing on PM2.5, S and N depositions, as well as ozone. There are a number of important results coming out of this study, including the contribution of ship-derived sulfate and nitrate to the aerosols, seasonality in coastal ammonium sulfate/nitrate related to agriculture-derived NH3, and the transition between O3 formation/titration as a function distance from emission and VOC availability. The source receptor relationship will be a useful reference for policy makers and other users. Model sensitivity studies are done on both a global and regional scale, which help to constrain the effects of non-linearity, grid resolution, and boundary conditions.

[Figure]

My only major complaint is that the discussion section seems to be on the light side. There is no comparison made between these modelling results with previous observational or model estimates. As such, the model results look to be qualitatively reasonable and sensible, but it's hard to know how quantitative they are.

Finally, while perhaps not the standard output of EMEP, it'd be interesting to explore some additional parameters that are also important for air pollution/atmospheric chemistry with the model, such as:

- Fraction of ship-derived PM that is secondary vs. primary as a function of distance from emission?

- Estimation of ship-derived PM1 and total aerosol number concentration?

A few minor edits:

Line 151. NMVOC instead of NMVOX

Line 184. What's the difference between NOx and ShipNOx? That ShipNOx doesn't participate in O3 chemistry, and only deposits? Assigning 50% of NOx to this channel seems like a very simplistic way of treating the non-linear nature of ship plume chemistry. Is this how terrestrial stack NOx emission gets treated also?

Further works on plume chemistry modeling include (Charlton-Perez et al. ACP, 9, 7505-7518, 2009; Song et al JGR, vol 108, D4, 2003)

Line 373. SECAs instead of NECAs?

---

## Referee Comment (RC2) · Huan Liu (Referee) · 27 May 2020

Emissions from shipping are large sources of air pollution and depositions of oxidized nitrogen and sulfur. This study investigates effects of global ship emissions for specifically European area. The topic is important for further understanding on air pollution mechanism and source-receptor relationship. Although the shipping emission estimation for CAP2020 is rough, the air quality modelling part was well designed and conducted. Results are interesting and most of which were explained in detail. So, this manuscript could meet the quality requirements of journal. However, following concerns should be addressed before acceptance.

Specific comments: 1. Globally the highest d(PM) concentrations are calculated over

parts of Asia and North Africa. Authors have not cited any regional results in East Asia or North Africa, or observation. A comparison on the results for regional level is necessary. At least, Fig 2b shows significant underestimation in China (please refer to "Impacts of shipping emissions on PM2.5 pollution in China. Atmospheric Chemistry and Physics 2018, 18, 15811-15824").

2. I don't understand the logic for ozone part, specifically Fig. 5. The ratio between NOx and NMVOC, which determine the ozone titration of formation, is for the receptor instead of source. How can Fig. 5 present opposite effects in the same receptor region, same season by different source regions' shipping emissions? Same problem for Fig. 6. The only explanation is that ozone was transported instead of the precursors. If so, the way authors discuss the issue in section 3.2 should be revised to focus on the ozone transportation. Then another question comes, if the ozone titration happens in where the emissions were released, does the model show ozone reduction in the emission region as well? Anyway, ozone transportation related analysis needs to be conducted. Current version could not provide full support for results.

3. SR (Source Receptor) calculations are not described very clearly. How to determine the 100% of the effects of source contribution by 15% reduced source emissions? If 10% or 20% were chose, will the source contribution (100%) be different?

4. Pseudo-species "ShipNOx" also needs further explanation. 50% as shipNOx, then how about other 50%? In this 50%, how much will go to R1 and how much for R2?

5. The connections with authors' previous paper of "Effects of strengthening the Baltic Sea ECA regulations" need to be built. Are the 2016 and 2017 results comparable in Baltic Sea region? Any differences? Connections on scenarios?

6. Line 214-217. The well-known mechanisms explained here were not connected to the specific findings in either text or figures. So what? Any nitrate ammonia peak in spring, or in summer?

7. No speciation was provided for PM2.5. Thus, the talking ammonia, nitrate or sulfate was not supported well, which also linked to question 6.

---

## Referee Comment (RC3) · Anonymous Referee #3 · 6 Jun 2020

This paper describes an interesting study regarding the comparison of global and regional numerical modeling results to evaluate the impact of shipping emissions on air quality over the globe (and in particular over Europe region). The work described is consistent, but there are some open questions and fragilities that should be solved and discussed before publications. The major problems are related to the absence of no model validation is presented or mentioned (model uncertainty should be presented and discussed), quantitative analysis should be always preferable instead of qualitative and it is not clear the added value of this paper comparing to other recent ones (refered in state-of-art) with very similar objectives and modeling (and scenarios) approaches. Below, major and minor revisions requested.

Abstract Line 4: The authors should be consistent when presenting the pollutants

[Figure]

(name or chemical compound). Please harmonize this along the text. Line 7: The objective/purpose of the study is missing in the abstract! Line 10: Something should be mentioned about the shipping emissions inventory used here (a particularly important input for this study) Lines 18-22: this conclusion is too general and obvious. There are more specific and interesting conclusions at the end of the paper that should be here mentioned.

Introduction Line 26: strange way of starting this Introductory section Line 30: land or maritime emissions? Line 44: reference should be added to support this Line 64-67: it is not clear which is the novelty of this study comparing to others recently published like for example Sofiev et al (2018). The authors should also explained why only focus on PM2.5 and ozone. Also the modeling system could be already mentioned/identified in this part.

Model description Lines 81-84: a reference is missing Line 144: which are higher: emissions per grid cell or total emissions? Line 151-152: please review this sentence Line 158: I do not understood this part "for several of the model runs"...please clarify.

Results (3-5 chapters) Line 218: how did the authors calculate this nitrate contribution? Line 253: Section 5 instead of Sections 5 Lines 271-272: Please review this sentence Line 296: ozone is, in particular, high... Lines 303-306: please quantify these contributions Line 322-323: please clarify/explain why these contributions are negative in these areas Line 328: please review this sentence Line 332: please quantify the ozone reductions mentioned Lines 334-342: the same comment before applies here (quantifications would be important) Line 366: please review this sentence Section 5.1/5.2: the authors identified previously a group of (significant) differences between the global and regional simulations (namely land and shipping emissions, scenarios applications, boundary and initial conditions) but they do not use these differences to explain some of the differences found in results. These differences, in particular, the emission data should be discuss - and in particular why these difference do bot invalidate the comparison between the simulations

Conclusions Line 462: I would suggest to modify the sentence to "Assuming the fulfillment of the legislation, it is expected that this result in substantial..." Lines 481, 487: please review these sentences
* * *

---

## Author Comment (AC3) · 2 Jul 2020

We, the authors, thank the reviewers for constructive comments and suggestions. Below we list the comments from reviewer 3, followed by our reply with references to changes made in the paper.

**Comments to remarks from reviewer 3**

Abstract Line 4:

The authors should be consistent when presenting the pollutants Please harmonize this along the text.

Reply:
*We have harmonized the naming of the emitted species.*

Line 7: The objective/purpose of the study is missing in the abstract!

Reply:
*In the abstract we have now added that we in this paper quantify the contributions from international shipping to European air pollution levels and depositions.*

Line 10: Something should be mentioned about the shipping emissions inventory used here (a particularly important input for this study).

Reply:
*We now state that the ship emissions have been derived using ship positioning data.*

Lines 18-22: this conclusion is too general and obvious. There are more specific and interesting conclusions at the end of the paper that should be here mentioned.

Reply:
*We have extended the abstract, including some more points from the conclusions.*

Introduction Line 26: strange way of starting this Introductory section.

Reply:
*Our intention is to point out that land based emissions in Europe have dropped significantly in past decades, whereas ship emissions have changed far less over the same period. Thus the relative contributions to air pollution and depositions have increased. This first paragraph has been slightly revised.*

Line 30: land or maritime emissions?
Reply:
*We have added land based emissions.*

Line 44: reference should be added to support this.
Reply:
*We have added a reference to the IMO decision in 2008.*

Line 64-67: It is not clear which is the novelty of this study comparing to others recently publishedlike for example Sofiev et al (2018). The authors should also explained why only focuson PM2.5 and ozone. Also the modeling system could be already mentioned/identified in this part.
Reply:
*In (line 84 - 99 in revised manuscript) we list the main topics discussed in the paper, clearly stating in which aspects provides added value beyond previous publications. Specifically, in addition to PM2.5 and ozone we also include depositions of oxidised nitrogen and sulphur. We also attributer the the effects of ship emissions from separate sea areas to specific European countries.*

Model description Lines 81-84: a reference is missing.
Reply:
*References were given lower on the same page. We have moved the references to a few lines below what was suggested by the reviewer.*

Line 144: which are higher:emissions per grid cell or total emissions?
Reply:
*We now specify that it is the total emissions in the sea areas that are higher in the global model.*

Line 151-152: please review this sentence.
Reply:
*We have changed this to:*
*However, as shown in Table 1, the NMVOC to $NO_x$ ratio is close to one for land based emissions, but very low for ship emissions.*

Line 158: I do not understood this part "for several of the model runs" ...please clarify.
Reply:
*We have clarified this point adding more text and referring to Table 2:*
*"The global model runs are made for a full calendar year (2017). As some of the species have a long lifetime in the atmosphere (one month or more), the model runs are preceded by a 5 months spin-up. But for model runs perturbing only a limited sea area, the spin-up from the Base model run is used (see Table 2)."*

Line 218: how did the authors calculate this nitrate contribution?
Reply: *Nitrate chemistry is included in the EMEP model. We have now included figure panels (in Figure 8) showing the fraction (of $PM_{2.5}$) of nitrate and also the fractions of*

*PPM and ammonium. ]*

*Line 253: Section 5 instead of Sections*
*Reply:*
*This is corrected.*

*Lines 271-272: Please review this sentence.*
*Reply:*
*We have split this part into several sentences, making it easier to understand.*

*Line 296: ozone is, in particular, high...*
*Reply:*
*We have added the missing word "production".*

*Lines 303-306: please quantify these contributions.*
*Reply:*
*We have added:*
*"This is shown in more detail for the country attributions section below. "*

*Line 322-323: please clarify/explain why these contributions are negative in these ar-*
*eas.*
*Answer:*
*We have added "as a result of ozone titration".*

*Line 328: please review this sentence.*
*Reply:*

*We have broken up this sentence to make it clearer.*

*Line 332: please quantify the ozonereductions mentioned.*
*REply:*
*We have rewritten this part, quantifying contributions.*

*Lines 334-342: the same comment before applies here (quantifications would be important).*
*Reply:*
*We have now quantified the depositions in the text.*

*Line 366: please review this sentence.*
*Reply:*
*We have rewritten this sentense to: "In Figure 4 the contributions to* $PM_{2.5}$ *from all ships to selected European countries are shown as a percentage of all anthropogenic contributions calculated with ship emissions before and after the implementation of CAP2020. "*

*Section 5.1/5.2:*
*the authors identified previously a group of (significant) differences between the global and regional simulations (namely land and shipping emissions, scenarios applications,boundary and initial conditions) but they do not use these differences to explain some of the differences found in results. These differences, in particular, the emission data should be discuss - and in particular why these difference do not invalidate the comparison between the simulations.*
*Reply:*
*We have added that for both* $PM_{2.5}$ *and the depositions of oxidised nitrogen and sul-*

*phur most of the difference is caused by the higher emissions used in the global model calculations.*

*Conclusions Line 462:*
*I would suggest to modify the sentence to "Assuming the fulfillment of the legislation, it is expected that this result in substantial..."*
*Reply:*
*We have changed the sentence as suggested.*

*Lines 481, 487:please review these sentences:*
*Reply:*
*We have changed this part of the paper to: "Thus the additional benefits of global model calculations are small compared to the improvements in accuracy that can be achieved with finer resolution on a smaller model domain. For ozone, enhancing the resolution improves the representation of localised variations in $NO_x$ to NMVOC ratios, explaining the differences in particular in the high $NO_x$ emitting countries and regions bordering the North Sea. On the other hand, with global scale calculations the contributions to ozone from all global sources can be included. For several countries/regions we show that for ozone, contributions from ROW shipping are comparable, and in some regions higher than, the contributions from sea areas close to Europe. "*

*Interactive comment on Atmos. Chem. Phys. Discuss., https://doi.org/10.5194/acp-2020-293, 2020.*

---

## Author Response (AR1)

Dear Editor

Below follows a point-by-point reply to the comments raised in the reviewers (answers to the comments in italics). We greatly appreciate the comments, as they have helped us improve the quality of the paper. There are also some additional minor changes in the manuscript in addition to those requested in the review. All changes are included in the marked up manuscript included below.

**Comments from reviewer 1 followed by our reply with references to changes made in the paper where needed.**

My only major complaint is that the discussion section seems to be on the light side.There is no comparison made between these modelling results with previous observational or model estimates. As such, the model results look to be qualitatively reasonable and sensible, but it's hard to know how quantitative they are.

Reply:
*The EMEP model has been compared to measurements as well as to other models. References to several of these studies are given in section 2. In order to make this clearer we have separated the model evaluation and model inter-comparisons into a new subsection. Here we have also added additional material, referring to Karl et al. (2019), comparing the EMEP model to the SILAM model and the CMAQ model as well as measurements. In this paper model calculated effects of ship emissions in the Baltic Sea are also compared.*

*With reference to this new sub-section (in particular see lines 142 - 155 in marked-up manuscript) these results are further discussed in the conclusions.*

Finally, while perhaps not the standard output of EMEP, it'd be interesting to explore some additional parameters that are also important for air pollution/atmospheric chem-istry with the model, such as:

Fraction of ship-derived PM that is secondary vs. primary as a function of distance from emission?

Reply:
*In addition to the fraction of sulphate in PM2.5 we now also include three additional figures (Figure 8 b,c,d) showing the fraction of nitrate, primary particles and ammonium in PM2.5. Note that sulphate emitted as primary particles (roughly 5% of the total sulphur emissions from ships) is included as sulphate and not primary particles.*

Estimation of ship-derived PM1 and total aerosol number concentration.

Reply:
*Unfortunately PM1 and particle number are not included in the EMEP model.*

**Minor edits**

Line 151
NMVOX corrected to NMVOC

Line 184.
What's the difference between NOx and ShipNOx? That ShipNOx doesn't participate in O3 chemistry, and only deposits? Assigning 50% of NOx to this channel seems like a very simplistic way of treating the non-linear nature of ship plume chemistry. Is this how terrestrial stack NOx emission gets treated also? Further works on plume chemistry modeling include (Charlton-Perez et al. ACP, 9,7505-7518, 2009; Song et al JGR, vol 108, D4, 2003)

Reply:
*The reviewer is correct in stating that this is a very simplistic way of treating the non-linear nature of ship plume chemistry. In particular in pristine environments the chemical regime in the ship plumes will be very different from the surrounding airmasses. In the parameterization applied in the Vinken et al. paper they calculate a strong ozone titration in the first stages of the plume, followed by ozone production as the plume expands. At this stage OH levels become higher than in the surroundings, resulting in a faster conversion of $NO_2$ to $HNO_3$, thus shortening the lifetime of NOx. The shorter lifetime of NOx and the increase of $HNO_3$ found by Vinken et al. is mimicked by the simplistic "SHIPNOX" parameterisation, removing NOx that would otherwise produce (too much) ozone and convert it directly to $HNO_3$. The parameterisation is included in order to give a range for the effects of ship emissions on ozone in otherwise pristine environments where there are no or few nearby sources. Terrestrial stacks are not (or very seldom) located in pristine environments, so we do not use a "SHIPNOX" type of parameterisation for these.*

Line 373
*NECA has been replaced by SECA*

**Comments from reviewer 2 followed by our reply with references to changes made in the paper where needed.**

Specific comments 1:
Globally the highest d(PM) concentrations are calculated over parts of Asia and North Africa. Authors have not cited any regional results in East Asia or North

Africa, or observation. A comparison on the results for regional level is necessary. At least, Fig 2b shows significant underestimation in China (please refer to: "Impacts of shipping emissions on PM2.5 pollution in China. Atmospheric Chemistry and Physics 2018, 18, 15811-15824)".

Reply:

*The contribution of ships to annual average PM2.5 in Lv et al (2018) is around 3-5$\mu gm^{-3}$ around Shanghai (Fig 3a of Lv (2018)). This is higher, but roughly comparable to our estimate of shipping contribution, which is 2-3$\mu gm^{-3}$ for the same area. The spatial patterns of 0.1 microgram per cubic meter concentration contour line from shipping are also similar. There can be several reasons for differences, e.g use of different emission inventories, atmospheric models and weather data. Regardless, both studies are fairly consistent of shipping contribution to PM2.5 concentration over this area. Several factors that can cause differences are listed below:*

- *First, our study uses STEAM3 emission inventories which are based on the modelling of individual ships and their engines. In STEAM, fuel type assignment is done based on engine characteristics which differs from the Liu et al approach, who seemed to assume OGVs always using HSHFO. This difference is likely to be small because most of the large ships operating in this area use 2-stroke main engines for propulsion, with the exception of cruise vessels which often rely on 4-stroke MSDs.*

- *The second obvious difference comes with the activity data. Lv et al use 2013 as a base year of shipping activity, whereas our study describes year 2017. It is likely that during this time vessel activity has increased because of a recovery of the economy after the 2008 financial crisis.*

- *A third difference is the activity source itself. The current work is based on the combination of satellite and terrestrial AIS data provided by Orbcomm Ltd, whereas Lv et al do not state their data source, but this is most likely higher update rate AIS from Chinese national network than the Orbcomm data used in this work.*

- *A fourth difference which could explain the lower shipping contribution in the Shanghai area is the inclusion of low sulphur emission zones (Pearl river Delta, Bohai Sea and Yangtze river delta) which is included in the STEAM3 inventories. Since Lv et al concentrate on 2013 emissions, this emission reduction did not exist at that time. This will cut PM emissions easily by 50%, reducing the PM total emissions from ships.*

- *The regional model calculations of Lv et al. used a finer model resolution than in our study (36 x 36 km versus 0.5 x 0.5 degrees). A finer resolution may result in higher peak concentrations.*

*In our view, our results of PM shipping contribution to air quality are in agreement with the earlier work of Lv et al. (2019) both in spatial pattern and in magnitude.*

*We have included a reference to the Lv et al. (2018) paper in the Introduction (line 84 - 88 marked-up paper):*
*" With ship emissions representative for year 2013, Lv et al. (2018) calculated contributions from ship emissions to $PM_{2.5}$ concentrations of up to 5.2 $\mu gm^{-3}$ in coastal regions of China, higher than in European coastal regions. Since 2013 emission controls have been imposed in China in several steps, limiting the fuel sulfur content in marine fuels to 0.5% in several Chinese ports and territorial waters."*

*And in the conclusions (lines 584 - 591 in marked-up manuscript):*
*"In Chinese coastal regions the peak contributions to $PM_{2.5}$ concentrations in this study are lower than in the study by ?. There are several possible explanations for this difference. Lv et al. (2018) used a finer model resolution (36 × 36 km) than in the present study. A finer resolution is likely to result in somewhat higher peak concentrations. Stricter regulations, limiting the sulfur content in marine fuels to 0.5% in and around several Chinese ports, including the YRD (Yangtze River Delta), have been imposed between these two studies (2013 versus 2017), and are included in the ECCAD 2017 ship emission data. According to Lv et al. (2018) YRD is responsible for about 20% of the ship emissions in Chinese waters. "*

Specific comments 2:
I don't understand the logic for ozone part, specifically Fig. 5. The ratio between NOx and NMVOC, which determine the ozone titration of formation, is for the receptor instead of source. How can Fig. 5 present opposite effects in the same receptor region, same season by different source regions' shipping emissions? Same problem for Fig.6. The only explanation is that ozone was transported instead of the precursors. If so, the way authors discuss the issue in section 3.2 should be revised to focus on the ozone transportation. Then another question comes, if the ozone titration happens in where the emissions were released, does the model show ozone reduction in the emission region as well? Anyway, ozone transportation related analysis needs to be conducted. Current version could not provide full support for results.
Reply:
*We now explain this in the first part of section 3. (lines 252 - 262 in the marked-up paper) as follows:*
*"Below we include the model results from all ship emissions, and from ship emissions in separate sea areas based on the model scenarios listed in Table 2, For the calculations perturbing the emissions in separate sea areas, the total effect in a receptor area will then be the sum of contributions from all the individual sea areas. This sum will then be a combination of the emission and chemical production/destruction of the species within the source sea area, and production/destruction of the species elsewhere (including the receptor region). Similar positive and negative contributions was also shown in the TF_HTAP2 model experiment, exemplified by the results in Jonson et al. (2019) and in the EMEP source receptor calculations, exemplified by EMEP, (2019), appendix C. Thus, for example, reductions in the receptor area can be caused by chemical reactions that only occur in the source area (e.g. ozone titration), followed by transport of a smaller amount of the species (e.g. ozone) into*

*the target area."*

Specific comments 3:
SR (Source Receptor) calculations are not described very clearly. How to determine the 100% of the effects of source contribution by 15% reduced source emissions? If 10% or 20% were chose, will the source contribution (100%) be different?
Reply:
*This is descussed in section 2.3. In Table 5 and 6 the depositions from shipping are scaled by 100/15. This scaling is made in Figure 2, right panes and in Figure 3 and 5.*

*Non-linearities are not a model uncertainty. The contribution of one source area depends on contributions from other source areas. This complication also applies to other source receptor methods, e.g. tagging methods or adjoint methods. Also in the real world, if one was able to switch off one source at a time (100% reduction), and do this for all sources, one after another (and measure each increment individually), then the sum of all individual increments would be different from the total concentration in the receptor area. This is due to the non-linearity of atmospheric chemistry. The choice of 15% was made for reasons explained in the manuscript. 'Contributions' calculated in this way should be seen as a measure of what can be achieved by emission reductions of this order of magnitude in the source areas.*
*In section 2.3 we now include the following (lines 224 - 228 in marked-up paper):*
*"Reducing the emissions by a different percentage would give different results depending on species and location. The choice of 15% is partially political as reductions of this magnitude are achievable within a timeframe of a few years and at the same time give a large enough signal when processing the model output."*

Specific comment 4:
Pseudo-species "ShipNOx" also needs further explanation. 50% as shipNOx, then how about other 50%? In this 50%, how much will go to R1 and how much for R2?
Reply;
*We now specify that the remaining 50% is emitted as NO and NO2 as in the Base model runs (for shipping 95% as NO). To trace how much will go through R1 and how much through R2 is not possible with the present model setup. See lines 236 - 237 in marked-up paper.*

Specific comment 5:
The connections with authors' previous paper of "Effects of strengthening the Baltic-Sea ECA regulations" need to be built. Are the 2016 and 2017 results comparable in Baltic Sea region? Any differences? Connections on scenarios.
Reply:
*We have included a reference to the 2019 paper in the introduction, and also to the companion paper Barregaard et al. (2019). See marked-up paper lines 75 - 88. In both the 2019 paper and the present paper we use the EMEP model, but with different model domain and model resolution. In both studies ship emissions from FMI and land based emissions from Eclipse are used. Model results are comparable in these two studies bearing mind that the meteorological year is different and that model perturbations are only made for the Baltic Sea.*

Specific comment 6: Line 214-217. The well-known mechanisms explained here were not connected to the specific findings in either text or figures. So what? Any nitrate ammonia peak in spring, or in summer?

Reply: *See reply to specific comment 7*

Specific comment 7:

No speciation was provided for PM2.5. Thus, the talking ammonia, nitrate or sulfate was not supported well, which also linked to question 6.

Reply:

*In addition to the figure showing the fraction of sulfate in PM2.5, we have now included figures showing the corresponding fractions of nitrate, primary particles and ammonium in Figure 8 b,c,d. As now mentioned in the paper, ship emissions will result in the formation of sodium nitrate particles, but these are generally large particles not contributing to PM2.5. Thus the seasonal variation of nitrate shown in Figure 3 is a result of ammonium nitrate.*

**Comments from reviewer 3 followed by our reply with references to changes made in the paper where needed.**

Abstract Line 4:

The authors should be consistent when presenting the pollutants Please harmonize this along the text.

Reply:

*We have harmonized the naming of the emitted species.*

Line 7: The objective/purpose of the study is missing in the abstract!

Reply:

*In the abstract we have now added that we in this paper quantify the contributions from international shipping to European air pollution levels and depositions.*

Line 10: Something should be mentioned about the shipping emissions inventory used here (a particularly important input for this study).

Reply:

*We now state that the ship emissions have been derived using ship positioning data.*

Lines 18-22: this conclusion is too general and obvious. There are more specific and interesting conclusions at the end of the paper that should be here mentioned.

Reply:

*We have extended the abstract, including some more points from the conclusions.
(see lines 23 - 35 in marked - up paper.)*

Introduction Line 26: strange way of starting this Introductory section.

Reply:

*Our intention is to point out that land based emissions in Europe have dropped sig-
nificantly in past decades, whereas ship emissions have changed far less over the
same period. Thus the relative contributions to air pollution and depositions have
increased. This first paragraph has been slightly revised.*

Line 30: land or maritime emissions?
Reply:

*We have added land based emissions. (line 40 marked - up paper).*

Line 44: reference should be added to support this.
Reply:

*We have added a reference to the IMO decision in 2008. (Line 56 in marked - up
paper).*

Line 64-67: It is not clear which is the novelty of this study comparing to others
recently publishedlike for example Sofiev et al (2018). The authors should also
explained why only focuson PM2.5 and ozone. Also the modeling system could be
already mentioned/identified in this part.
Reply:

*In (line 89 - 104 in marked - up paper) we list the main topics discussed in the paper,
clearly stating in which aspects provides added value beyond previous publications.
Specifically, in addition to PM2.5 and ozone we also include depositions of oxidised
nitrogen and sulphur. We also attributer the the effects of ship emissions from
separate sea areas to specific European countries.*

Model description Lines 81-84: a reference is missing.
Reply:

*References were given lower on the same page. We have moved the references to a
few lines below what was suggested by the reviewer. (Lines 114 - 117 in marked -
up paper).*

Line 144: which are higher:emissions per grid cell or total emissions?
Reply:

*We now specify that it is the total emissions in the sea areas that are higher in the
global model. (Line 187 in marked - up paper).*

Line 151-152: please review this sentence.
Reply:

*We have changed this to:
However, as shown in Table 1, the NMVOC to $NO_x$ ratio is close to one for land*

*based emissions, but very low for ship emissions. (lines 197 - 198 in marked up paper).*

Line 158: I do not understood this part "for several of the model runs" ...please clarify.
Reply:
*We have clarified this point adding more text and referring to Table 2:*
*"The global model runs are made for a full calendar year (2017). As some of the species have a long lifetime in the atmosphere (one month or more), the model runs are preceded by a 5 months spin-up. But for model runs perturbing only a limited sea area, the spin-up from the Base model run is used (see Table 2)." (lines 206 - 207 in marked - up paper).*

Line 218: how did the authors calculate this nitrate contribution?
Reply: *Nitrate chemistry is included in the EMEP model. We have now included figure panels (in Figure 8) showing the fraction (of* $PM_{2.5}$*) of nitrate and also the fractions of PPM and ammonium. ]*

*Line 253: Section 5 instead of Sections*
*Reply:*
*This is corrected.*

*Lines 271-272: Please review this sentence.*
*Reply:*
*We have split this part into several sentences, making it easier to understand. (lines 346 - 351 in marked - up paper).*

*Line 296: ozone is, in particular, high...*
*Reply:*
*We have added the missing word "production".*

*Lines 303-306: please quantify these contributions.*
*Reply:*
*We have added:*
*"This is shown in more detail for the country attributions section below. " We have also included more information in section 3.2.5 (see lines 404 - 418 in marked up paper).*

*Line 322-323: please clarify/explain why these contributions are negative in these areas.*
*Answer:*
*We have added "as a result of ozone titration".*

*Line 328: please review this sentence.*
*Reply:*
*We have broken up this sentence to make it clearer.*

*Line 332: please quantify the ozonereductions mentioned.*
*REply:*
*We have rewritten this part, quantifying contributions.*

*Lines 334-342: the same comment before applies here (quantifications would be important).*
*Reply:*
*We have now quantified the depositions in the text. (see lines 427 - 432 in marked - up paper).*

*Line 366: please review this sentence.*
*Reply:*
*We have rewritten this sentense to: "In Figure 4 the contributions to $PM_{2.5}$ from all ships to selected European countries are shown as a percentage of all anthropogenic contributions calculated with ship emissions before and after the implementation of CAP2020. " (lines 461 - 463 in marked - up paper).*

*Section 5.1/5.2:*
*the authors identified previously a group of (significant) differences between the global and regional simulations (namely land and shipping emissions, scenarios applications,boundary and initial conditions) but they do not use these differences to explain some of the differences found in results. These differences, in particular, the emission data should be discuss - and in particular why these difference do not invalidate the comparison between the simulations.*
*Reply:*
*We have added that for both $PM_{2.5}$ and the depositions of oxidised nitrogen and sulphur most of the difference is caused by the higher emissions used in the global model calculations. (lines 501 - 503 and lines 521 - 522 in marked - up paper).*

*Conclusions Line 462:*
*I would suggest to modify the sentence to "Assuming the fulfillment of the legislation, it is expected that this result in substantial..."*
*Reply:*
*We have changed the sentence as suggested. (see lines 568 - 569 in marked - up paper).*

*Lines 481, 487:please review these sentences:*
*Reply:*
*We have made significant changes in ther conclusion section, including the sentence in question.*

[revised manuscript text omitted]

**a)** Annual average $PM_{2.5}$ in $\mu gm^{-3}$

**b)** Annual average $PM_{2.5}$ from ships in $\mu gm^{-3}$

**c)** Annual ozone in ppb

**d)** Annual average ozone from ships in ppb

**e)** Annual dep. oxidised N in $mgNm^{-2}$

**f)** Annual dep. oxidised N in $mgNm^{-2}$ from shipping

**g)** Annual dep. oxidised S in $mgSm^{-2}$

**h)** Annual dep. oxidised S in $mgSm^{-2}$ from shipping

**Figure 2.** Right: Annually averaged global concentrations of $PM_{2.5}$ a) and $O_3$ c). Depositions of oxidised nitrogen e) and sulfur g). Left: Contributions from global shipping to $PM_{2.5}$ b) and $O_3$ d) and to depositions of oxidised nitrogen f) and sulfur h). The contributions from shipping have been multiplied by 100/15.

[Figure]

**Figure 3.** Seasonal contributions to European $PM_{2.5}$ levels (in $\mu gm^{-3}$) from 15% perturbations of the emissions in separate sea areas defined in section 2.3.  Winter defined as December–January, Spring: March–May, Summer:June–August, Autumn: September–November. The contributions from shipping have been multiplied by 100/15.

[Figure]

**Figure 4.** Percentage contributions from shipping to annually averaged $PM_{2.5}$ to countries bordering the North Sea and the Baltic Sea (left) and the Mediterranean Sea and the Black Sea (right) relative to contributions from all global anthropogenic emissions. Contributions are shown both for all ships and separated by sea area. For each country the contributions from the individual sea areas are added in the upper bar and the contributions from all ship emissions calculated as the difference between the Base and SR_AllShips scenarios are shown as black + grey bar below. The Base - SR_AllShips bars are split in a black and grey part where the first grey part represents the contributions after CAP2020 and black + grey the contributions prior to CAP2020. Differences in length between All ships (Black + grey) and the added contributions from the separate sea areas is an indication of non-linear effects.

[Figure]

**Figure 5.** Seasonal contributions to European ozone levels (in ppb) from 15% perturbations of the emissions in separate sea areas defined in section 2.3.  Winter defined as December–January, Spring: March–May, Summer:June–August, Autumn: September–November. The contributions from shipping have been multiplied by 100/15.

[Figure]

**Figure 6.** Contributions from shipping in ppb to annually averaged ozone from 15% reductions in ship emissions (left). Numbers to the right of the country names are the effects of the 15% reductions of all antropogenic emissions calculated as Base_2017 – SR_AllSh. Right, percentage contributions to SOMO35 relative to contributions from all global anthropogenic emissions. Contributions are shown for all ships and separated by sea area. The length of the bars are split so that the darker parts of the bars represent calculations assuming SHIPNOX (see section 2) and the full length without SHIPNOX. Note that for Malta the smaller perturbation in $NO_x$ from Mediterranean shipping with SHIPNOX results in a small ppb increase in calculated ozone, whereas the larger perturbation without SHIPNOX results in a decrease.

[Figure]

**Figure 7.** Percentage contributions from shipping to annually averaged depositions of oxidised nitrogen (top) and sulfur (bottom) relative to contributions from all global anthropogenic emissions. Contributions are shown for all ships and separated by sea area. see also caption in Figure 4.

[Figure]

**a)** Fraction of $SO_4$ in $PM_{2.5}$ from ships

**b)** Fraction of nitrate in $PM_{2.5}$ from ships

**c)** Fraction of PPM in $PM_{2.5}$ from ships

**d)** Fraction of ammonium in $PM_{2.5}$ from ships

**e)** Reductions in $PM_{2.5}$ ($\mu gm^{-3}$) following CAP2020

**Figure 8.** Fraction of a) $SO_4$, b) nitrate, c) PPM and d) ammonia in $PM_{2.5}$ in European waters from shipping a).  PPM are Ash, EC and OC. e) reductions in $PM_{2.5}$ ($\mu gm^{-3}$) following the CAP2020 regulations.